# Stability of motor representations after paralysis

**Charles Guan[1]\*[†], Tyson Aflalo[1,2]\*[†], Carey Y Zhang[1], Elena Amoruso[3], Emily R Rosario[4], Nader Pouratian[5], Richard A Andersen[1,2]**

[1]California Institute of Technology, Pasadena, United States; [2]T&C Chen Brain-Machine Interface Center at Caltech, Pasadena, United States; [3]Institute of Cognitive Neuroscience, University College London, London, United Kingdom; [4]Casa Colina Hospital and Centers for Healthcare, Pomona, United States; [5]UT Southwestern Medical Center, Dallas, United States

**Abstract** Neural plasticity allows us to learn skills and incorporate new experiences. What happens when our lived experiences fundamentally change, such as after a severe injury? To address this question, we analyzed intracortical population activity in the posterior parietal cortex (PPC) of a tetraplegic adult as she controlled a virtual hand through a brain–computer interface (BCI). By attempting to move her fingers, she could accurately drive the corresponding virtual fingers. Neural activity during finger movements exhibited robust representational structure similar to fMRI recordings of able-bodied individuals' motor cortex, which is known to reflect able-bodied usage patterns. The finger representational structure was consistent throughout multiple sessions, even though the structure contributed to BCI decoding errors. Within individual BCI movements, the representational structure was dynamic, first resembling muscle activation patterns and then resembling the anticipated sensory consequences. Our results reveal that motor representations in PPC reflect able-bodied motor usage patterns even after paralysis, and BCIs can re-engage these stable representations to restore lost motor functions.

**\*For correspondence:**
cguan@caltech.edu (CG);
taflalo@caltech.edu (TA)

[†]These authors contributed equally to this work

**Competing interest:** The authors declare that no competing interests exist.

## Editor's evaluation

Using data from an tetraplegic individual, the authors show that the neural representations for attempted single finger movements after multiple years after the injury is still organized in a way that is typical for healthy participants. They also show that the representational structure does not change during task training on a simple finger classification task – and that the representational structure, even without active motor outflow or sensory inflow, switches from a motor representation to a sensory representation during the trial. The analyses are convincing, and the results have important implications for the use and training of BCI devices in humans.

## Introduction

A central question in neuroscience is how experience affects the nervous system. Studies of this phenomenon, plasticity, were pioneered by Hubel and Wiesel, who found that temporary visual occlusion in kittens can induce lifelong reorganization of the visual cortex (*Hubel and Wiesel, 1970*). Their results demonstrated that the developing brain, rather than being genetically preprogrammed, is surprisingly malleable to external inputs.

Subsequent studies showed that other brain regions are also plastic during early development, but it is unclear how plastic the nervous system remains into adulthood. Visual occlusion in adult cats does not reorganize the visual cortex, and lesion studies of the adult visual cortex have arrived

at competing conclusions of reorganization and stability (*Smirnakis et al., 2005*; *Gilbert and Wiesel, 1992*; *Keck et al., 2008*; *Baseler et al., 2011*). A similar discussion continues regarding the primary somatosensory cortex (S1). Classical studies posited that amputation and spinal cord injury modify the topography of body parts in S1, with intact body parts taking over cortical areas originally dedicated to the amputated part (*Merzenich et al., 1984*; *Qi et al., 2000*; *Pons et al., 1991*; *Jain et al., 2008*). However, recent human neuroimaging studies and sensory BCI studies have challenged the extent of this remapping, arguing that sensory topographies largely persist even after complete sensory loss (*Makin and Bensmaia, 2017*; *Kikkert et al., 2021*; *Flesher et al., 2016*; *Armenta Salas et al., 2018*). Thus, the level of plasticity in the adult nervous system is still an ongoing investigation.

Understanding plasticity is necessary to develop brain–computer interfaces (BCIs) that can restore sensorimotor function to paralyzed individuals (*Orsborn et al., 2014*). First, paralysis disrupts movement and blocks somatosensory inputs to motor areas, which could cause neural reorganization (*Pons et al., 1991*; *Jain et al., 2008*; *Kambi et al., 2014*). Second, BCIs bypass supporting cortical, subcortical, and spinal circuits, fundamentally altering how the cortex affects movement. Do these changes require paralyzed BCI users to learn fundamentally new motor skills (*Sadtler et al., 2014*), or do paralyzed participants use a preserved, pre-injury motor repertoire (*Hwang et al., 2013*)? Several paralyzed participants have been able to control BCI cursors by attempting arm or hand movements (*Hochberg et al., 2006*; *Hochberg et al., 2012*; *Collinger et al., 2013*; *Gilja et al., 2015*; *Bouton et al., 2016*; *Ajiboye et al., 2017*; *Brandman et al., 2018*), hinting that motor representations could remain stable after paralysis. However, the nervous system's capacity for reorganization (*Pons et al., 1991*; *Jain et al., 2008*; *Kikkert et al., 2021*; *Kambi et al., 2014*) still leaves many BCI studies speculating whether their findings in tetraplegic individuals also generalize to able-bodied individuals (*Flesher et al., 2016*; *Armenta Salas et al., 2018*; *Stavisky et al., 2019*; *Willett et al., 2020*; *Fifer et al., 2022*). A direct comparison, between BCI control and able-bodied neural control of movement, would help address questions about generalization and plasticity.

Temporal dynamics provide another lens to investigate neural organization and its changes after paralysis. Temporal signatures can improve BCI classification (*Willett et al., 2021*) or provide a baseline for motor adaptation studies (*Stavisky et al., 2017*; *Vyas et al., 2018*). Notably, motor cortex (MC) activity exhibits quasi-oscillatory dynamics during arm reaching (*Churchland et al., 2012*). More generally, the temporal structure can depend on the movement type (*Suresh et al., 2020*) and the recorded brain region (*Schaffelhofer and Scherberger, 2016*). In this study, we recorded from the posterior parietal cortex (PPC), which is thought to compute an internal forward model for sensorimotor control (*Mulliken et al., 2008*; *Wolpert et al., 1998*; *Desmurget and Grafton, 2000*; *Li et al., 2022*). A forward-model overcomes inherent sensory delays to enable fast control by predicting the upcoming states. If PPC activity resembles a forward model after paralysis, this would suggest that even the temporal details of movement are preserved after injury.

Here, we investigate the neural representational structure of BCI finger movements in a tetraplegic participant. In able-bodied individuals, the cortical organization of finger representations follows the natural statistics of movements (*Ejaz et al., 2015*; *Lillicrap and Scott, 2013*). In a BCI task, the experimenter can instruct movement patterns unrelated to biomechanics or before-injury motifs. In this study, we tested whether the neural representational structure of BCI finger movements by a tetraplegic individual matches that of able-bodied individuals performing similar, overt movements, or whether the structure follows the task's optimal representational structure (*Bonnasse-Gahot and Nadal, 2008*). If the BCI finger organization matches that of able-bodied movement, participants would likely be able to activate pre-injury motor representations, indicating that motor representations were preserved after paralysis.

We report that the neural representational structure of BCI finger movements in a tetraplegic individual matches that of able-bodied individuals. This match was stable across sessions, even though the measured representational structure contributed to errors in the BCI task. Furthermore, the neural representational dynamics matched the temporal profile expected of a forward model, first resembling muscle activation patterns and then resembling expected sensory outcomes. Our results suggest that adult motor representations in PPC remain even after years without use.

# Results

## Intracortical recordings during finger flexion

We recorded single and multineuron activity (95.8 ± standard deviation [SD] 6.7 neurons per session over 10 sessions) from participant NS while she attempted to move individual fingers of the right hand. We recorded from a microelectrode array implanted in the left (contralateral) PPC at the junction of the postcentral and intraparietal sulci (PC-IP, *Figure 1—figure supplement 1*). This region is thought to specialize in the planning and monitoring of grasping movements (*Andersen et al., 2019*; *Orban and Caruana, 2014*; *Gallivan and Culham, 2015*; *Klaes et al., 2015*).

Each recording session started with an initial calibration task (*Figure 1—figure supplement 2*, Methods). On each trial, we displayed a text cue (e.g., 'T' for thumb) on a computer screen, and the participant immediately attempted to flex the corresponding finger, as though pressing a key on a keyboard. Because participant NS previously suffered a C3–C4 spinal cord injury resulting in tetraplegia (AIS-A), her movement attempts did not generate overt motion. Instead, participant NS attempted to move her fingers as though she was not paralyzed.

These attempted movements resulted in distinct neural activity patterns across the electrode array. To enable BCI control, we trained a linear classifier (Methods) to identify finger movements from neural firing rates. The participant subsequently performed several rounds of a similar finger flexion task, except that (1) the trained classifier now provided text feedback of its predicted finger and (2) the task randomized the visual cue location (*Figure 1a* and Methods). We repeated this online-control finger flexion task over multiple sessions (408 ± SD 40.8 trials/session over 10 sessions) and used this data for our offline analyses. Participant NS also performed a control task, identical in structure except that she attended to cues without performing the corresponding movements.

## Accurately decoding fingers from PPC single-neuron activity

High classification accuracy during online control (86% ± SD 4% over 10 sessions; chance = 17%) (*Figure 1b* and *Figure 1—figure supplement 3*) and offline cross-validated classification (92% ± SD 2%; *Figure 1—figure supplement 4a*) demonstrated that the finger representations were reliable and linearly separable. During the calibration task, cross-validated classification was similarly robust (accuracy = 96% ± SD 3%; *Figure 1—figure supplement 4b*). These finger representations were robust across contexts and could be used in a range of environments, including to move the hand of a virtual reality avatar (*Video 1*).

At the single-neuron level, most (89%) neurons were significantly tuned to individual finger-press movements (significance threshold: p < 0.05, FDR-corrected) (*Figure 1—figure supplement 5*). *Figure 1c–f* show the firing rates of example neurons, which were tuned to one or more fingers and change tuning profiles over the course of each movement.

To confirm that the observed neural responses could not be explained by visual confounds, we verified that we could not discriminate between fingers during the control task (*Figure 1—figure supplement 6*). Furthermore, we could not decode the gaze location during the finger classification time window in the standard online-control task (*Figure 1—figure supplement 6*). Thus, reliable finger representations emerged from the participant's movement attempts.

## Finger representational structure matches the structure of able-bodied individuals

Having discovered that PC-IP neurons modulate selectively for finger movements, we next investigated how these neural representations were functionally organized and how this structure related to pre-injury movements. Here, we turned to the framework of representational similarity analysis (RSA) (*Kriegeskorte et al., 2008a*; *Diedrichsen and Kriegeskorte, 2017*). RSA quantifies neural representational structure by the pairwise distances between each finger's neural activity patterns (*Figure 2a*). These pairwise distances form the representational dissimilarity matrix (RDM), a summary of the representational structure. Importantly, these distances are independent of the original feature types (e.g., electrode or voxel measurements), allowing us to compare finger organizations across subjects and across recording modalities (*Kriegeskorte et al., 2008b*).

We used RSA to test three hypotheses: (1) the BCI finger representational structure could match that of able-bodied individuals (*Ejaz et al., 2015*; *Kieliba et al., 2021*; *Figure 2b* and *Figure 2—figure*

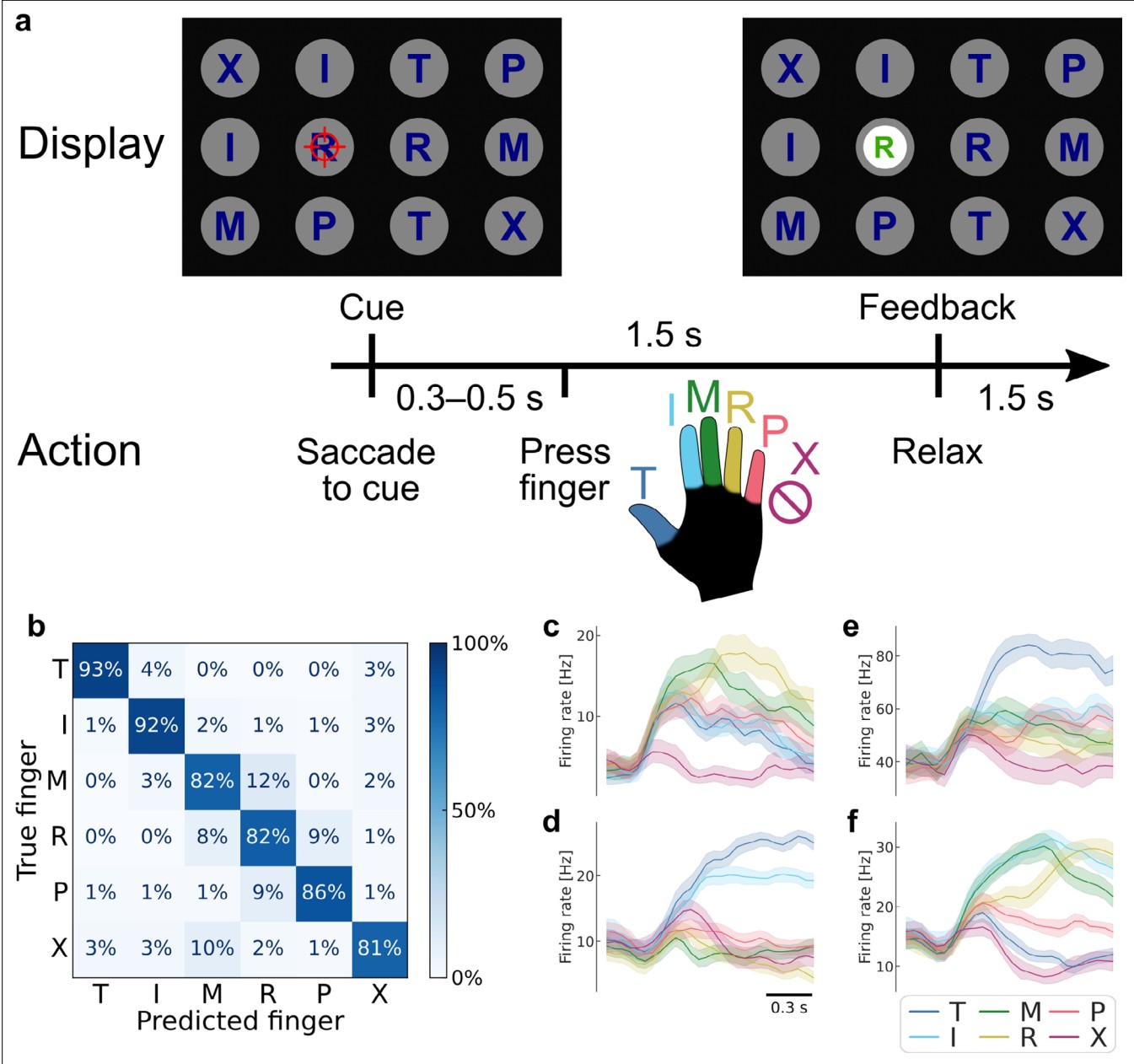

**Figure 1.** Robust brain–computer interface (BCI) control of individual fingers. (**a**) Main finger flexion task. When a letter was cued by the red crosshair, the participant looked at the cue and immediately attempted to flex the corresponding finger of her right (contralateral) hand. We included a null condition 'X', during which the participant looked at the target but did not move her fingers. Visual feedback indicated the decoded finger 1.5 s after cue presentation. To randomize the gaze location, cues were located on a grid (three rows, four columns) in a pseudorandom order. The red crosshair was jittered to minimize visual occlusion. (**b**) Confusion matrix showing robust BCI finger control (86% overall accuracy, 4016 trials aggregated over 10 sessions). Each entry (*i*, *j*) in the matrix corresponds to the ratio of movement *i* trials that were classified as movement *j*. (**c–f**) Mean firing rates for four example neurons, color-coded by attempted finger movement. Shaded areas indicate 95% confidence intervals (across trials of one session). Gaussian smoothing kernel (50 ms standard deviation [SD]).

The online version of this article includes the following figure supplement(s) for figure 1:

**Figure supplement 1.** Multielectrode array implant location.

**Figure supplement 2.** Calibration task.

**Figure supplement 3.** Brain–computer interface (BCI) classification accuracy across sessions.

**Figure supplement 4.** Robust cross-validated finger classification during main and calibration tasks.

**Figure supplement 5.** Single-neuron encoding of individual fingers.

**Figure supplement 6.** Gaze location did not affect finger decoding during the attempted-movement period.

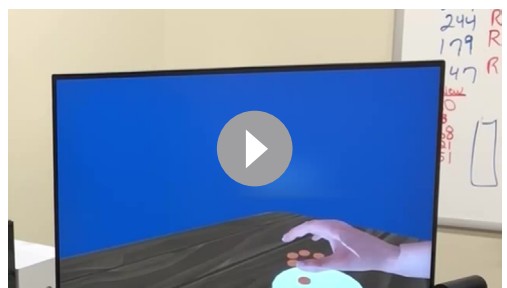

**Video 1.** Example brain–computer interface (BCI) control of a virtual reality hand. Using a BCI, participant NS controls the individual fingers of a virtual reality hand. She views a virtual hand, table, and cues through an Oculus headset. Similar to the main finger movement task, she acquires green jewels by pressing the corresponding finger and avoids red amethysts by resting. Green jewels disappear when the correct finger is classified (or at the start of the next trial, if incorrectly classified). The screen copies the view that participant NS sees through the Oculus headset.

https://elifesciences.org/articles/74478/figures#video1

supplement 1), which would imply that motor representations did not reorganize after paralysis. This hypothesis would be consistent with recent functional magnetic resonance imaging (fMRI) studies of amputees, which showed that sensorimotor cortex representations of phantom limb finger movements match the same organization found in able-bodied individuals (*Kikkert et al., 2016*; *Wesselink et al., 2019*). We note that our able-bodied model was recorded from human PC-IP using fMRI, which measures fundamentally different features (millimeter-scale blood oxygenation) than microelectrode arrays (sparse sampling of single neurons). Another possibility is that (2) the participant's pre-injury motor representations had despecialized after paralysis, such that finger activity patterns are unstructured and pairwise independent (*Figure 2c*). However, this hypothesis would be inconsistent with results from fMRI studies of amputees' sensorimotor cortex (*Kikkert et al., 2016*; *Wesselink et al., 2019*). Lastly, (3) the finger movement representational structure might optimize for the statistics of the task (*Lillicrap and Scott, 2013*; *Clancy et al., 2014*). Our BCI task, as well as previous experiments with participant NS, involved no correlation between individual fingers, so the optimal structure would represent each finger independently to minimize confusion between fingers. In other words, the task-statistics hypothesis (3) would predict that, with BCI usage, the representational structure would converge toward the task-optimal, unstructured representational structure (*Figure 2c*).

Does the finger representational structure in a tetraplegic individual match that of able-bodied individuals? We quantified the finger representational structure by measuring the cross-validated Mahalanobis distance (Methods) between each finger pair, using the firing rates from the same time window used for BCI control. The resulting RDMs are shown in *Figure 2d* (average across sessions) and *Figure 2—figure supplement 2* (all sessions). For visual intuition, we also projected the representational structure to two dimensions in *Figure 2e*, which shows that the thumb is distinct, while the middle, ring, and pinky fingers are close in neural space. We then compared the measured RDMs against the able-bodied fMRI and unstructured models, using the whitened unbiased RDM cosine similarity (WUC) (*Diedrichsen et al., 2021*). The measured representational structure matched the able-bodied representational structure significantly over the unstructured model ($p = 5.7 \times 10^{-5}$, two-tailed $t$-test) (*Figure 2f*), ruling out the despecialization hypothesis (2). Our findings were robust to different choices of distance and model-similarity metrics (*Figure 2—figure supplement 3*).

We note that we constructed the able-bodied fMRI model from the mean of PC-IP fMRI RDMs across multiple able-bodied participants ($N = 29$). When compared among the RDM distribution of individual able-bodied participants, participant NS's average PC-IP RDM was statistically typical (permutation shuffle test, $p = 0.55$), in part because PC-IP fMRI RDMs were relatively variable across able-bodied participants (*Figure 2—figure supplement 4*).

We also compared the PC-IP BCI RDM with able-bodied fMRI MC RDMs, which have been previously shown to match the patterns of natural hand use (*Ejaz et al., 2015*). The able-bodied MC and PC-IP fMRI finger organizations are similar in that they represent the thumb distinctly from the other fingers, but PC-IP fMRI signals represent each of the non-thumb fingers similarly while MC distinguishes between all five fingers (*Figure 2—figure supplement 1*). Interestingly, PC-IP BCI finger representations matched the able-bodied fMRI finger representational structure in the MC (*Figure 2—figure supplement 1*) even better than that of able-bodied PC-IP (*Figure 2—figure supplement 5*). The WUC similarity with the MC RDM was close to the noise ceiling (Methods), indicating that the MC RDM matches participant NS's data better than almost any other model could (see Discussion).

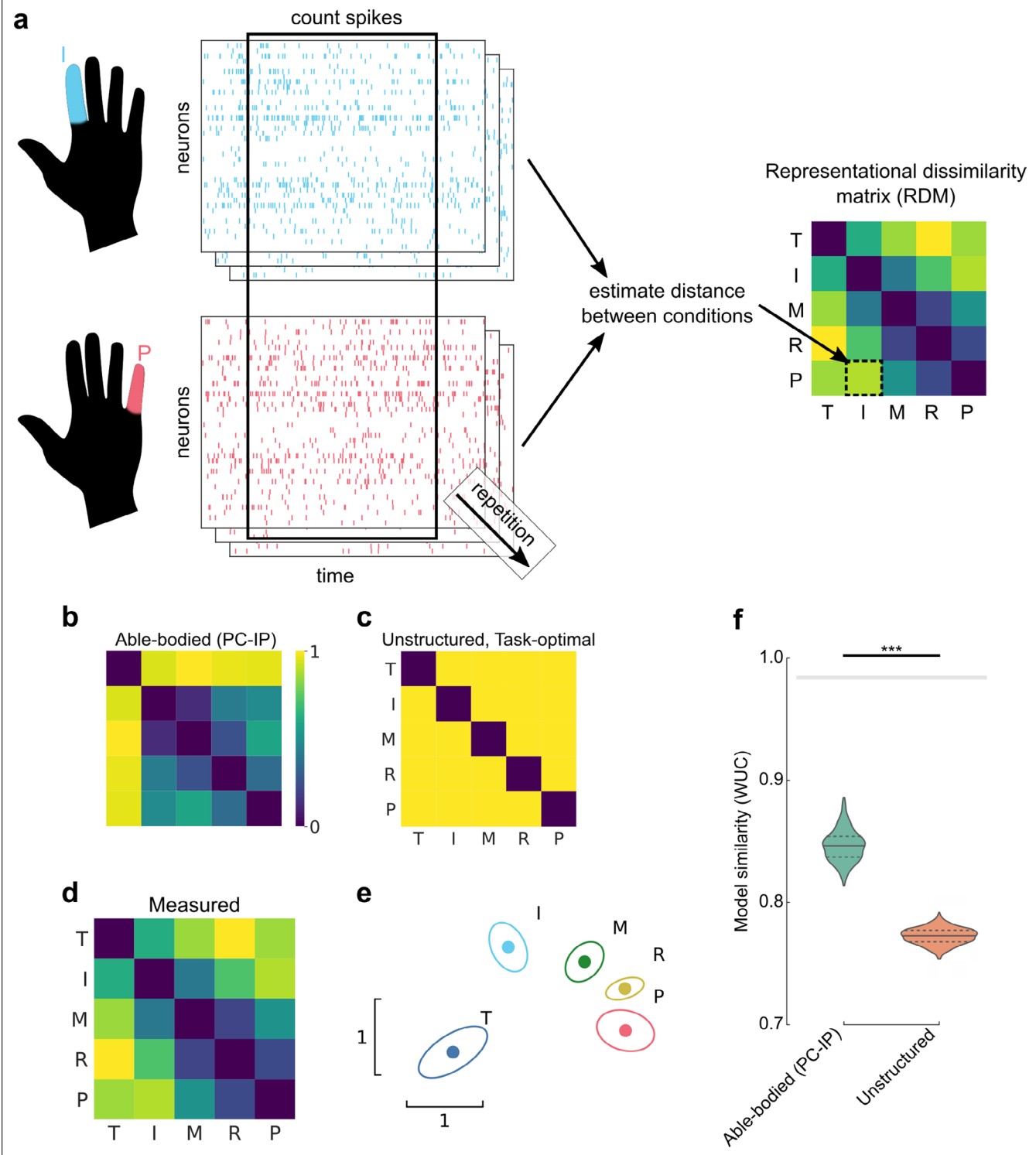

**Figure 2.** Representational structure during brain–computer interface (BCI) finger control matches the structure of able-bodied individuals. (**a**) To calculate the representational dissimilarity matrix (RDM), a vector of firing rates was constructed for each trial. Repetitions were collected for each condition. Then, pairwise distances were estimated between conditions using a cross-validated dissimilarity metric. This process was repeated to generate an RDM for each session. We drop the No-Go condition (X) here to match previous finger studies (*Ejaz et al., 2015*; *Kikkert et al., 2016*). (**b**) Representational structure hypothesized by the preserved-representation hypothesis: average RDM for 36 able-bodied individuals performing a finger-press task. RDMs were measured at the junction of the postcentral and intraparietal sulci (PC-IP) using fMRI (*Ejaz et al., 2015*; *Kieliba et al., 2021*). Max-scaled to [0, 1]. (**c**) Representational structure hypothesized by the despecialization and task-optimal hypotheses: pairwise-equidistant RDM.

*Figure 2 continued on next page*

*Figure 2 continued*

Max-scaled to [0, 1]. (**d**) Finger representational structure measured in tetraplegic participant NS: cross-validated Mahalanobis distances (Methods) between neural activity patterns, averaged across 10 recording sessions. Max-scaled to [0, 1]. (**e**) Intuitive visualization of the distances in (**d**) using multidimensional scaling (MDS). Ellipses show mean ± standard deviation (SD) (10 sessions) after Generalized Procrustes alignment (without scaling) across sessions. (**f**) Measured RDMs (**d**) match the able-bodied PC-IP fMRI RDM (**b**) better than they match the task-optimal, unstructured model (**c**), as measured by the whitened unbiased cosine similarity (*Diedrichsen et al., 2021*) (WUC) (Methods). Mean differences were significant (able-bodied vs. unstructured, p = $5.7 \times 10^{-5}$; two-tailed *t*-test, 1000 bootstrap samples over 10 sessions). Violin plot: solid horizontal lines indicate the median WUC over bootstrap samples, and dotted lines indicate the first and third quartiles. Noise ceiling: gray region estimates the best possible model fit (Methods). Asterisks denote a significant difference at ***p < 0.001. For convenience, a similar figure using a correlation-based similarity metric is shown in *Figure 2—figure supplement 3*.

The online version of this article includes the following figure supplement(s) for figure 2:

**Figure supplement 1.** fMRI representational structure for finger movements, from *Kieliba et al., 2021*.

**Figure supplement 2.** Individual representational dissimilarity matrices (RDMs) for each session.

**Figure supplement 3.** Representational structure during brain–computer interface (BCI) finger control matches the structure of able-bodied individuals when using alternative analysis parameters.

**Figure supplement 4.** fMRI finger representational dissimilarity matrices (RDMs) are more consistent across able-bodied participants in motor cortex (MC) than in the junction of postcentral and intraparietal sulci (PC-IP).

**Figure supplement 5.** Finger representational structure of the tetraplegic individual, measured at the junction of the postcentral and intraparietal sulci (PC-IP), matches fMRI representational dissimilarity matrices (RDMs) from motor cortex (MC) even better than fMRI RDMs from PC-IP.

## Representational structure did not trend toward task optimum

Next, we investigated whether the BCI finger representational structure matched that of able-bodied individuals consistently or whether the representational structure changed over time to improve BCI performance. The task-optimal structure hypothesis (3) predicted that the BCI RDMs would trend to optimize for the task statistics (unstructured model, *Figure 2c*) as the participant gained experience with the BCI task. However, we did not find conclusive evidence for a trend from the able-bodied model toward the unstructured model (linear-model session × model interaction: *t*(6) = 0.50, one-tailed *t*-test p = 0.32, Bayes factor [BF] = 0.66) (*Figure 3a*). Indeed, participant NS's finger RDMs were largely consistent across different recording sessions (average pairwise correlation, excluding the diagonal: *r* = 0.90 ± SD 0.04, min 0.83, max 0.99).

We considered whether learning, across sessions or within sessions, could have caused smaller-scale changes in the representational structure. The observed representational structure, where middle-ring and ring-pinky pairs had relatively small distances, was detrimental to classification performance. The majority (70%) of the online classification errors were middle-ring or ring-pinky confusions (*Figure 1b*). Due to these systematic errors, one might reasonably predict that plasticity mechanisms would improve control by increasing the inter-finger distances between the confused finger pairs. Contrary to this prediction, the middle-ring and ring-pinky distances did not increase over the course of the experiment (across sessions: *t*(8) = −4.5, one-tailed *t*-test p > 0.99, BF = 0.03; across runs within sessions: *t*(82) = −0.45, one-tailed *t*-test p = 0.67, BF = 0.12) (*Figure 3b*). When analyzing all finger pairs together, the inter-finger distances also did not increase (across sessions: *t*(8) = −4.0, one-tailed *t*-test p = 0.98, BF = 0.01; across runs within sessions: *t*(74) = −2.4, one-tailed *t*-test p = 0.99, BF = 0.02), as visualized by the similarity between the average early-half RDM and the average late-half RDM (*Figure 3c*). These analyses demonstrate that the representational structure did not trend toward the task optimum (*Figure 2c*) with experience, ruling out the task-statistics hypothesis (3).

## Finger representational structure is motor-like and then somatotopic

PPC is hypothesized to overcome inherent sensory delays by computing an internal forward model for rapid sensorimotor control (*Mulliken et al., 2008*; *Wolpert et al., 1998*; *Desmurget and Grafton, 2000*.) The forward model integrates an efference copy of motor signals and delayed sensory feedback to dynamically predict the state of the body. The hypothesized forward-model role would predict that the representational structure changes over the time course of each movement, with an early motor-command-like component during movement initiation. To investigate this temporal evolution, we modeled the representational structure of finger movements at each timepoint as a non-negative linear combination (*Kietzmann et al., 2019*) of potentially predictive models (*Figure 4a*).

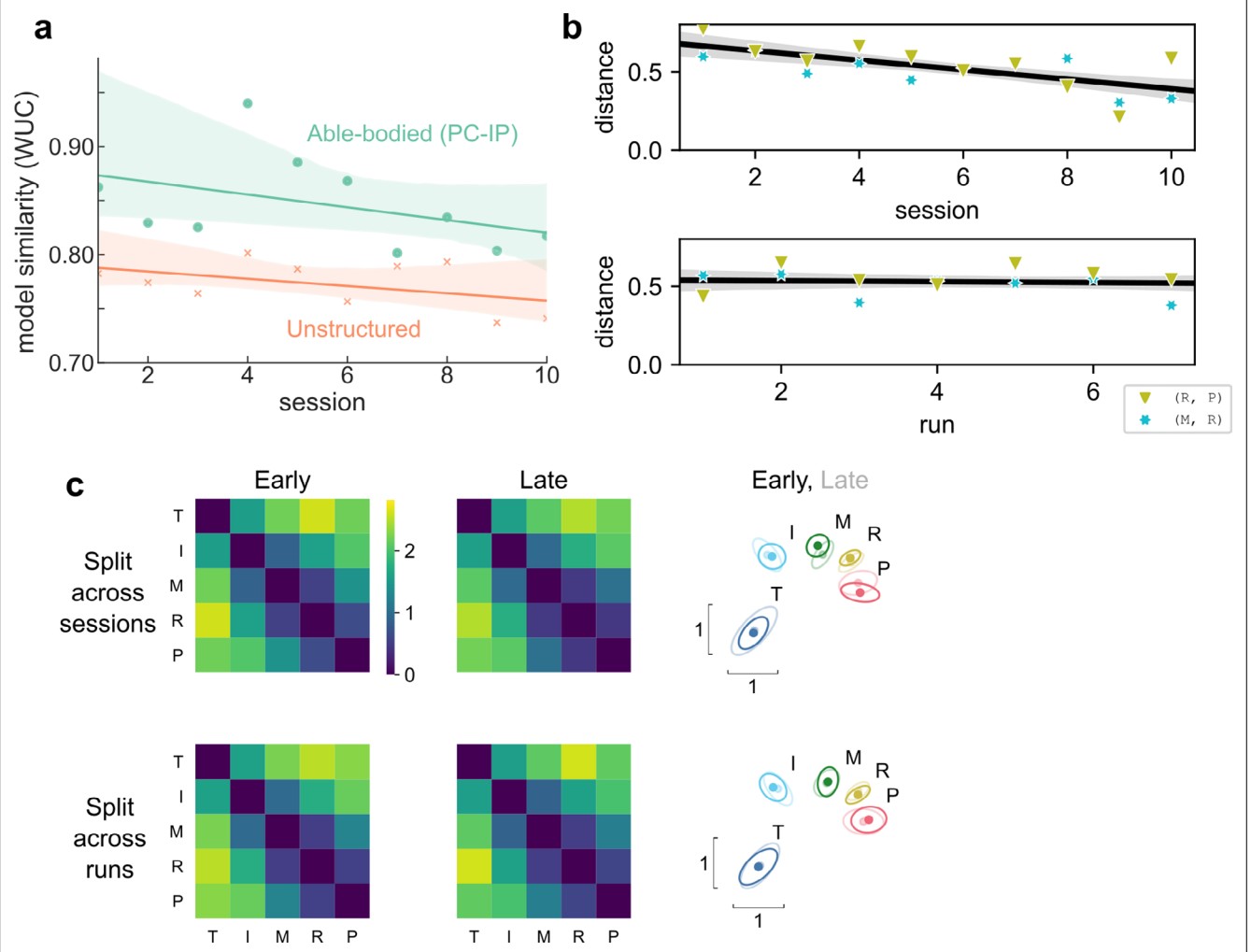

**Figure 3.** Hand representation changed minimally after weeks of brain–computer interface (BCI) control. (**a**) Slope comparison shows that the model fit did not trend toward the unstructured model over sessions (p = 0.32). (**b**) The distance between high-error finger pairs (middle-ring and ring-pinky) did not increase across sessions or runs (within sessions), as shown by partial regression plots. Distance metric: cross-validated Mahalanobis, averaged across runs (for the session plot) or averaged across sessions (for the run plot). The black line indicates linear regression. The gray shaded region indicates a 95% confidence interval. Each run consisted of 8 presses per finger. (**c**) Minimal change in representational structure between early and late sessions or between early and late runs. Mean representational dissimilarity matrix (RDM), when grouped by sessions (top row) or individual runs (bottom row). Grouped into early half (left column) or late half (center column). Multidimensional scaling (MDS) visualization (right column) of early (opaque) and late (translucent) representational structures after Generalized Procrustes alignment (without scaling, to allow distance comparisons).

The online version of this article includes the following figure supplement(s) for figure 3:

**Figure supplement 1.** Inter-finger distances did not increase across sessions or within sessions.

We considered three models (**Ejaz et al., 2015**) that could account for representational structure: hand usage, muscle activation, and somatotopy. The hand-usage model (**Figure 4b**) predicts that the neural representational structure should follow the correlation pattern of finger kinematics during natural hand use. The muscle activation model (**Figure 4c**) predicts that the representational structure should follow the coactivation patterns of muscle activity during individual finger movements. The somatotopy model (**Figure 4d**) predicts that the representational structure should follow the spatial layout of the body, with neighboring fingers represented similarly to each other (**Ejaz et al., 2015**; **Schellekens et al., 2018**).**Schellekens et al., 2018** While somatotopy usually refers to physical spaces that resemble the body, here we use the term broadly to describe encoding spaces that resemble the body.

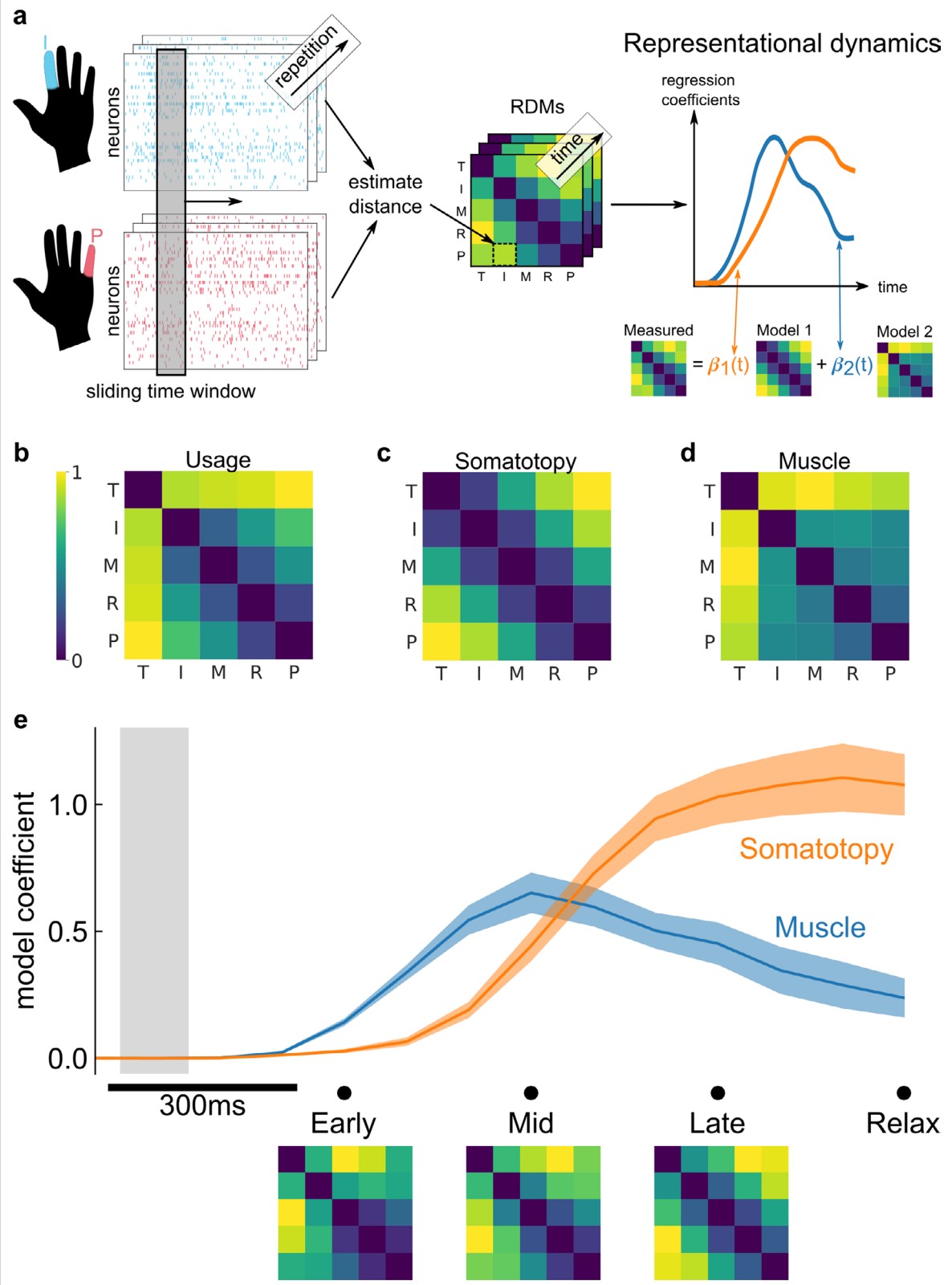

**Figure 4.** Representational dynamics analysis (RDA) dissociates neural processes over time. (**a**) RDA performs representational similarity analysis (RSA) in a sliding window across time. Here, we model the measured representational structure as a non-negative linear combination of component model representational dissimilarity matrices (RDMs). (**b–d**) Hypothesized explanatory component RDMs: usage, muscle, and somatotopy (*Ejaz et al., 2015*). Max-scaled to [0, 1]. (**e**) RDA of the measured RDM over time shows an early fit to the muscle model and a late fit to the somatotopy model. Confidence

*Figure 4 continued on next page*

*Figure 4 continued*
intervals indicate ± standard error of the mean (SEM) bootstrapped across 10 sessions. Gray shaded region indicates the approximate onset time of the saccade to cue (interquartile range across trials). Difference in model start time (170 ms, Methods) was significant (p = 0.002, two-sided Wilcoxon signed-rank test). RDM snapshots (bottom, each max-scaled to [0, 1]) intuitively visualize the change in representational structure over time from muscle-like to somatotopic.

The online version of this article includes the following figure supplement(s) for figure 4:

**Figure supplement 1.** Fit between measured representational dissimilarity matrix (RDM) and linear combinations of models.

**Figure supplement 2.** Temporal delays between component models are consistent across single sessions.

**Figure supplement 3.** Representational dynamics are robust across tasks and model combination choices.

**Figure supplement 4.** Well-isolated single neurons of the tetraplegic participant match the finger representational structure of able-bodied individuals.

Because the hand-usage model is nearly multicollinear with the muscle and somatotopy models (variance inflation factor: $VIF_{usage,OLS} = VIF_{usage,NNLS} = 20.9$, Methods), we first reduced the number of component models. Through a model selection procedure (Methods), we found that the hand usage + somatotopy and muscle + somatotopy model combinations matched the data best (*Figure 4—figure supplement 1*), with the muscle + somatotopy model matching the data marginally better. Thus, in the main text, we present our temporal analysis using the muscle and somatotopy component models.

*Figure 4e* shows the decomposition of the representational structure into the muscle and somatotopy component models. The results show a dynamic structure, with the muscle model emerging 170 ms earlier than the somatotopy model (p = 0.002, two-sided Wilcoxon signed-rank test). This timing difference was consistent across individual sessions (*Figure 4—figure supplement 2*) and task contexts, such as the calibration task (*Figure 4—figure supplement 3*). Indeed, the transition from the muscle model (*Figure 4c*) to the somatotopy model (*Figure 4d*) is visually apparent when comparing the average RDMs at the early (muscle-model-like) late (somatotopic) phases of movement (*Figure 4e*).

These temporal dynamics were robust to our model selection procedure, demonstrating a similar timing difference for the hand usage + somatotopy model combination (*Figure 4—figure supplement 3*).

## Discussion

### Neural prosthetic control of individual fingers using recordings from PC-IP

We found that participant NS could robustly control the movement of individual fingers using a neural prosthetic in a variety of contexts (*Figure 1*, *Figure 1—figure supplement 4*, *Video 1*), even after years of paralysis. Her BCI control accuracy exceeded the previous best of other five-finger, online BCI control studies (*Hotson et al., 2016*; *Jorge et al., 2020*). These results establish PC-IP as a candidate implant region for dexterous neural prostheses.

### Connecting BCI studies to basic neuroscience

Although previous studies have shown that the anterior intraparietal (AIP) area of PPC is involved in whole-hand grasping (*Schaffelhofer and Scherberger, 2016*; *Klaes et al., 2015*; *Murata et al., 2000*), our work is the first to discover individual finger representations in PPC (*Figure 1—figure supplement 5*). Likewise, many other BCI studies with tetraplegic participants have contributed to basic neuroscience, deepening our understanding of the human cortex (*Stavisky et al., 2019*; *Willett et al., 2020*; *Rutishauser et al., 2018*; *Zhang et al., 2017*; *Aflalo et al., 2020*; *Chivukula et al., 2021*). A frequent (*Flesher et al., 2016*; *Armenta Salas et al., 2018*; *Stavisky et al., 2019*; *Willett et al., 2020*; *Fifer et al., 2022*; *Chivukula et al., 2021*; *Andersen and Aflalo, 2022*) discussion question is: how well do these findings generalize to the brains of able-bodied individuals? Specifically, do the observed phenomena result from partial reorganization (*Kambi et al., 2014*; *Nardone et al., 2013*) after spinal cord injury, or do they reflect intact motor circuits, preserved from before injury (*Makin and Bensmaia, 2017*)?

Early human BCI studies (*Hochberg et al., 2006*; *Collinger et al., 2013*) recorded from the MC and found that single-neuron directional tuning is qualitatively similar to that of able-bodied non-human primates (NHPs) (*Hochberg et al., 2006*; *Georgopoulos et al., 1982*). Many subsequent human BCI studies have also successfully replicated results from other classical NHP neurophysiology studies (*Hochberg et al., 2012*; *Collinger et al., 2013*; *Gilja et al., 2015*; *Bouton et al., 2016*; *Ajiboye et al., 2017*; *Brandman et al., 2018*; *Aflalo et al., 2015*), leading to the general heuristic that the sensorimotor cortex retains its major properties after spinal cord injury (*Andersen and Aflalo, 2022*). This heuristic further suggests that BCI studies of tetraplegic individuals should generalize to able-bodied individuals. However, this generalization hypothesis had lacked direct, quantitative comparisons between tetraplegic and able-bodied individuals. Thus, as human BCI studies expanded beyond replicating results and beganto challenge conventional wisdom, neuroscientists questioned whether cortical reorganization could influence these novel phenomena (see Discussions of *Flesher et al., 2016*; *Armenta Salas et al., 2018*; *Stavisky et al., 2019*; *Willett et al., 2020*; *Fifer et al., 2022*; *Chivukula et al., 2021*; *Andersen and Aflalo, 2022*). As an example of a novel discovery, a recent BCI study found that the hand knob of tetraplegic individuals is directionally tuned to movements of the entire body (*Willett et al., 2020*), challenging the traditional notion that primary somatosensory and motor subregions respond selectively to individual body parts (*Penfield and Boldrey, 1937*). Given the brain's capacity for reorganization (*Jain et al., 2008*; *Kambi et al., 2014*), could these BCI results be specific to cortical remapping? Detailed comparisons with able-bodied individuals, as shown here, help shed light on this question.

## Matching finger organization between tetraplegic and able-bodied participants

We asked whether participant NS's BCI finger representations resembled that of able-bodied individuals or whether her finger representations had reorganized after paralysis. Single-neuron recordings of PC-IP during individuated finger movements for able bodied humans are not available for comparison. However, many fMRI studies have characterized finger representations (*Kikkert et al., 2021*; *Ejaz et al., 2015*; *Kikkert et al., 2016*; *Yousry et al., 1997*), and RSA has previously shown RDM correspondence between fMRI and single-neuron recordings of another cortical region (inferior temporal cortex) (*Kriegeskorte et al., 2008b*). This match was surprising because single-neuron and fMRI recordings differ fundamentally; single-neuron recordings sparsely sample $10^2$ neurons in a small region, while fMRI samples $10^4$–$10^6$ neurons/voxel (*Kriegeskorte and Diedrichsen, 2016*; *Guest and Love, 2017*). The correspondence suggested that RSA might identify modality-invariant neural organizations (*Kriegeskorte et al., 2008b*), so here we used fMRI recordings of human PC-IP as an able-bodied model.

We found that participant NS exhibited a consistent finger representational structure across sessions, and this representational structure matched the able-bodied fMRI model better than the task-optimal, unstructured model (*Figure 2*). When compared with individual able-bodied participants, participant NS's finger organization was also quite typical, in part due to the relative variability in PC-IP fMRI representational structure across able-bodied participants.

The MC fMRI finger representation is well studied and has been shown to reflect the patterns of natural hand use (*Ejaz et al., 2015*; *Kikkert et al., 2016*; *Wesselink et al., 2019*), so we also considered a model constructed from MC fMRI recordings. Compared to the PC-IP fMRI finger representation, MC represents the non-thumb fingers more distinctly from each other (*Figure 2—figure supplement 1*). Interestingly, participant NS's finger RDMs more strongly matched the able-bodied MC fMRI model, reaching similarities close to the theoretical maximum (*Figure 2—figure supplement 3* and *Figure 2—figure supplement 5*). This result does obscure a straightforward interpretation of the RSA results – why does our recording area match MC better than the corresponding implant location? Several factors might contribute, including differing neurovascular sensitivity to the early and late phases of the neural response (*Figure 4e*), heterogeneous neural organizations across the single-neuron and voxel spatial scales (*Kriegeskorte and Diedrichsen, 2016*; *Guest and Love, 2017*; *Arbuckle et al., 2020*), or mismatches in functional anatomy between participant NS and standard atlases (*Eickhoff et al., 2018*). Furthermore, fMRI BOLD contrast is thought to reflect cortical inputs and intracortical processing (*Logothetis et al., 2001*). Thus, the match between PC-IP spiking output and MC fMRI signals could also suggest that PC-IP sends signals to MC, thereby driving the observed MC fMRI structure.

Even so, it is striking that participant NS's finger representation matches the neural and hand use patterns (*Figure 4b* and *Figure 4—figure supplement 1*) of able-bodied individuals. Despite the lack of overt movement or biomechanical constraints (*Lang and Schieber, 2004*), the measured finger representation still reflected these usage-related patterns. This result matches recent sensorimotor cortex studies of tetraplegic individuals, where MC decoding errors (*Jorge et al., 2020*) and S1 finger somatotopy (*Kikkert et al., 2021*) appeared to reflect able-bodied usage patterns. Taken together with our dynamics analyses (see Discussion), the evidence supports the interpretation that motor representations are preserved after paralysis. Comparisons with single-neuron recordings from able-bodied participants would validate this interpretation. although such recordings may be difficult to acquire.

## Able-bodied-like finger representation is not explained by learning

Hand use patterns shape neural finger organization (*Ejaz et al., 2015*; *Kikkert et al., 2016*; *Wesselink et al., 2019*), so we also considered the possibility that participant NS's able-bodied-like representational structure emerged from BCI usage patterns after paralysis. Contrary to this hypothesis, her BCI finger representational structure changed minimally over weeks (*Figure 3*). Furthermore, even though participant NS's representational structure contributed to BCI errors (*Figure 1b*) and she was anecdotally cognizant of which fingers would get confused, she did not increase the neural distance between fingers with experience. This relative stability suggests that the measured representational structure has been stable after paralysis, rather than emergent from BCI learning.

The stability of finger representations here suggests that BCIs can benefit from the pre-existing, natural repertoire (*Hwang et al., 2013*), although learning can play an important role under different experimental constraints. In our study, the participant received only a delayed, discrete feedback signal after classification (*Figure 1a*). Because we were interested in understanding participant NS's natural finger representation, we did not artificially perturb the BCI mapping. When given continuous feedback, however, participants in previous BCI studies could learn to adapt to within-manifold perturbations to the BCI mapping (*Sadtler et al., 2014*; *Vyas et al., 2018*; *Ganguly and Carmena, 2009*; *Sakellaridi et al., 2019*). BCI users could even slowly learn to generate off-manifold neural activity patterns when the BCI decoder perturbations were incremental (*Oby et al., 2019*). Notably, learning was inconsistent when perturbations were sudden, indicating that learning is sensitive to specific training procedures.

To further understand how much finger representations can be actively modified, future studies could benefit from perturbations (*Kieliba et al., 2021*; *Oby et al., 2019*), continuous neurofeedback (*Vyas et al., 2018*; *Ganguly and Carmena, 2009*; *Oby et al., 2019*), and additional participants. Additionally, given that finger representations were dynamic (*Figure 4e*), learning could occur separately in the early and late dynamic phases. Time-variant BCI decoding algorithms, such as recurrent neural networks (*Willett et al., 2021*; *Sussillo et al., 2012*), could also help facilitate learning specific to different time windows of finger movement.

## Representational dynamics are consistent with PPC as a forward model

In able-bodied individuals, PPC is thought to maintain a forward estimate of movement state (*Mulliken et al., 2008*; *Wolpert et al., 1998*; *Desmurget and Grafton, 2000*; *Aflalo et al., 2015*; *McNamee and Wolpert, 2019*). As such, PPC receives delayed multimodal sensory feedback and is hypothesized to receive efference copies of motor command signals (*Mulliken et al., 2008*; *Andersen et al., 1997*). This hypothesized role predicts that PPC houses multiple functional representations, each engaged at different timepoints of motor production.

To dissociate these neural processes, we performed a time-resolved version of RSA (*Figure 4*). We considered three component models: muscle, usage, and somatotopy (*Ejaz et al., 2015*). Our temporal analysis showed a consistent ordering: early emergence of the muscle model followed by the somatotopy model.

This ordering was consistent when exchanging the muscle and hand-usage component models (*Figure 4* and *Figure 4—figure supplement 3*), as hand-usage and muscle activation patterns are strongly correlated for individual finger movements (*Overduin et al., 2012*). Therefore, we group these two models under the single concept of motor production. In the future, more complex multi-finger movements (*Ejaz et al., 2015*) would help distinguish between muscle and hand-usage models.

The somatotopy model predicts that neighboring fingers will have similar cortical activity patterns (*Ejaz et al., 2015*), analogous to overlapping Gaussian receptive fields (*Schellekens et al., 2018*). Gaussian receptive fields have been useful tools for understanding finger topographies within the sensorimotor cortex (*Schellekens et al., 2018*; *Schellekens et al., 2021*). In another study with participant NS, we found that the same PC-IP population encodes touch (*Chivukula et al., 2021*) with Gaussian-like receptive fields. Based on these results, the somatotopy model can be thought of as a sensory-consequence model. However, because participant NS has no sensation below her shoulders, we interpret the somatotopy model as the preserved prediction of the sensory consequences of a finger movement. These sensory outcome signals could be the consequence of internal computations within the PPC or could come from other structures important for body-state estimation, such as the cerebellum (*McNamee and Wolpert, 2019*).

The 170 ms timing difference we found roughly matches the delay between feedforward muscle activation and somatosensory afferents (*Scott, 2016*; *Sollmann et al., 2017*) in able-bodied individuals. Given PPC's hypothesized role as a forward model, PPC likely integrates motor planning and production signals to predict sensory outcomes at such a timing (*Mulliken et al., 2008*; *Wolpert et al., 1998*; *Desmurget and Grafton, 2000*; *McNamee and Wolpert, 2019*). Alternatively, because participant NS cannot move overtly, the sensory-consequence model could also reflect the error between the internal model's expected sensory outcomes and the actual (lack of) sensory feedback (*Adams et al., 2013*). In either scenario, the match in timing between BCI control and able-bodied individuals provides further evidence that the recorded motor circuits have preserved their functional role.

## Stability of sensorimotor representations after paralysis

A persistent question in neuroscience has been how experience shapes the brain, and to what extent existing neural circuits can be modified. Early studies by Merzenich et al. showed that the primary somatosensory cortex reorganized after amputation, with intact body parts invading the deprived cortex (*Merzenich et al., 1984*; *Qi et al., 2000*; *Pons et al., 1991*). However, the authors also recognized that the amputated body part might persist in latent somatosensory maps. Since then, preserved, latent somatosensory representations have been demonstrated in studies of amputation (*Makin and Bensmaia, 2017*; *Kikkert et al., 2016*; *Wesselink et al., 2019*; *Bruurmijn et al., 2017*) and even paralysis (*Kikkert et al., 2021*; *Flesher et al., 2016*; *Armenta Salas et al., 2018*; *Fifer et al., 2022*). Overall, deafferentation appears to expand the remaining regions slightly, even while the pre-injury structure persists in the deafferented cortex (*Makin and Bensmaia, 2017*). Fewer studies have investigated sensorimotor plasticity beyond the primary somatosensory cortex and MC, but our results in PC-IP indicate that association areas can also remain stable after paralysis.

The topic of cortical reorganization has long been significant to the development of BCIs, particularly when deciding where to implant recording electrodes. If, as previously thought, sensory deprivation drives cortical reorganization and any group of neurons can learn to control a prosthetic (*Fetz, 1969*; *Moritz and Fetz, 2011*), the specific implant location would not affect BCI performance. However, our results and others (*Smirnakis et al., 2005*; *Makin and Bensmaia, 2017*; *Kikkert et al., 2021*; *Hwang et al., 2013*; *Kikkert et al., 2016*; *Wesselink et al., 2019*; *Bruurmijn et al., 2017*) suggest that the pre-injury properties of brain regions do affect BCI performance. Even though experience shapes neural organization (*Merzenich et al., 1984*; *Ejaz et al., 2015*; *Wesselink et al., 2019*), representations may be remarkably persistent once formed (*Kikkert et al., 2021*; *Wesselink et al., 2019*). Thus, even though BCIs bypass limbs and their biomechanical constraints (*Lang and Schieber, 2004*), BCIs may still benefit from tapping into the preserved, natural (*Hwang et al., 2013*) movement repertoire of motor areas.

As BCIs enable more complex motor skills, such as handwriting (*Willett et al., 2021*), future studies could investigate whether these complex skills also retain their pre-injury representational structure. For example, does a tetraplegic participant's BCI handwriting look like their physical, pre-injury handwriting? These results will have important implications for the design of future neural prosthetics.

## Materials and methods

### Data collection

#### Study participant

The study participant NS has an AIS-A spinal cord injury at cervical level C3–C4 that she sustained approximately 10 years before this study. Participant NS cannot move or feel her hands. As part of a BCI clinical study (ClinicalTrials.gov identifier: NCT01958086), participant NS was implanted with two 96-channel Neuroport Utah electrode arrays (Blackrock Microsystems model numbers 4382 and 4383). She consented to the surgical procedure as well as to the subsequent clinical studies after understanding their nature, objectives, and potential risks. All procedures were approved by the California Institute of Technology, Casa Colina Hospital and Centers for Healthcare, and the University of California, Los Angeles Institutional Review Boards.

#### Multielectrode array implant location

The recording array was implanted over the hand/limb region of the left PPC at the junction of the intraparietal sulcus with the postcentral sulcus (*Figure 1—figure supplement 1*; Talairach coordinates [−36 lateral, 48 posterior, 53 superior]). We previously (*Klaes et al., 2015*; *Zhang et al., 2017*; *Aflalo et al., 2015*) referred to this brain area as the AIP area, a region functionally defined in NHPs. Here, we describe the implanted area anatomically, denoting it the PC-IP area. More details regarding the methodology for functional localization and implantation can be found in *Aflalo et al., 2015*.

#### Neural data preprocessing

Using the NeuroPort system (Blackrock Microsystems), neural signals were recorded from the electrode array, amplified, analog bandpass-filtered (0.3 Hz to 7.5 kHz), and digitized (30 kHz, 250 nV resolution). A digital high-pass filter (250 Hz) was then applied to each electrode.

Threshold crossings were detected at a threshold of −3.5× RMS (root-mean-square of an electrode's voltage time series). Threshold crossings were used as features for in-session BCI control. For all other analyses, we used $k$-medoids clustering on each electrode to spike-sort the threshold crossing waveforms. The first $n \in \{2, 3, 4\}$ principal components were used as input features to $k$-medoids, where $n$ was selected for each electrode to account for 95% of waveform variance. The gap criteria (*Tibshirani et al., 2001*) were used to determine the number of waveform clusters for each electrode.

### Experimental setup

#### Recording sessions

Experiments were conducted in 2–3 hr recording sessions at Casa Colina Hospital and Centers for Healthcare. All tasks were performed with participant NS seated in her motorized wheelchair with her hands resting prone on the armrests. Participant NS viewed text cues on a 27-inch LCD monitor that occupied approximately 40 degrees of visual angle. Cues were presented using the psychophysics toolbox (*Brainard, 1997*) for MATLAB (Mathworks).

The data were collected on 9 days over 6 weeks. Almost all experiment days were treated as individual sessions (i.e., the day's recordings were spike-sorted together). The second experiment day (2018-09-17) was an exception, with data being recorded in a morning period and an afternoon period with a sizable rest in between. To reduce the effects of recording drift, we treated the two periods as separate sessions (i.e., spike-sorted each separately) for a total of 10 sessions. Each session can thus be considered a different resampling of a larger underlying neural population, with both unique and shared neurons each session. We did not rerun the calibration task for the afternoon session of the second experiment day (2018-09-17), resulting in nine sessions of the calibration task for *Figure 1—figure supplement 4b*.

Each session consisted of a series of 2–3 min, uninterrupted 'runs' of the task. The participant rested for a few minutes between runs as needed.

#### Calibration task

At the beginning of each recording day, the participant performed a reaction-time finger flexion task (*Figure 1—figure supplement 2*; denoted 'calibration task' in the Results) to train a finger classifier for subsequent runs of the primary task. On each trial, a letter appeared on the screen

(e.g., 'T' for thumb). The participant was instructed to immediately flex the corresponding finger on the right hand (contralateral to the implant), as though pressing a key on a keyboard. The condition order was block-randomized, such that each condition appeared once before repetition.

### Finger flexion grid task

In the primary task, movement cues were arranged in a 3 × 4 grid of letters on the screen (*Figure 1a*). Each screen consisted of two repetitions each of T (thumb), I (index), M (middle), R (ring), P (pinky/ little), and X (No-Go) arranged randomly on the grid. Each trial lasted 3 s. At the beginning of each trial, a new cue was randomly selected with a crosshairs indicator, which jittered randomly to prevent letter occlusion. Each cue was selected once (for a total of 12 trials) before the screen was updated to a new arrangement. Each run consisted of three to four screens.

On each trial, the participant was instructed to immediately (1) saccade to the cued target, (2) fixate, and (3) attempt to press the corresponding finger. During both movement and No-Go trials, the participant was instructed to fixate on the target at least until the visual classification feedback was shown. The cue location randomization was used to investigate whether cue location would affect movement representations.

On each trial, 1.5 s after cue presentation, the classifier decoded the finger movement and presented its prediction via text feedback overlaid on the cue.

### No-movement control task

The control task was similar to the primary task, except that the subject was instructed to saccade to each cued letter and fixate without attempting any finger movements. No classification feedback was shown.

## Statistical analysis

### Unit selection

Single-unit neurons were identified using the *k*-medoids clustering method, as described in the Neural Data Preprocessing section. Analyses in the main text used all identified units, regardless of sort quality. With spike-sorting, there is always the possibility that a single waveform cluster corresponds to activity from multiple neurons. To confirm that potential multiunit clustering did not bias our results, we repeated our analyses using only well-isolated units (*Figure 4—figure supplement 4*).

Well-isolated single units were identified using the *L*-ratio metric (*Schmitzer-Torbert et al., 2005*). The neurons corresponding to the lowest third of *L*-ratio values (across all sessions) were selected as 'well-isolated'. This corresponded to a threshold of $L_{ratio} = 10^{-1.1}$ dividing well-isolated single units and potential multiunits (*Figure 4—figure supplement 4*).

### Single-unit tuning to finger flexion

We calculated the firing rate for each neuron in the window [0.5, 1.5] s after cue presentation. To calculate significance for each neuron (*Figure 1—figure supplement 5*), we used a two-tailed *t*-test comparing each movement's firing rate to the No-Go firing rate. A neuron was considered significantly tuned to a movement if $p < 0.05$ (after FDR correction). We also computed the mean firing rate change between each movement and the No-Go condition. If a neuron was significantly tuned to at least one finger, we denoted the neuron's 'best finger' as the significant finger with the largest effect size (mean firing rate change). For each neuron and finger, we also calculated the discriminability index (*d'*, RMS SD) between the baseline (No-Go) firing rate and the firing rate during finger movement.

In *Figure 1—figure supplement 5*, neurons were pooled across all 10 sessions. Neurons with mean firing rates less than 0.1 Hz were excluded to minimize sensitivity to discrete spike counting.

### Finger classification

To classify finger movements from firing rate vectors, we used linear discriminant analysis (LDA) with diagonal covariance matrices (*Dudoit et al., 2002*) (a form of regularization); diagonal LDA is also equivalent to Gaussian Naive Bayes (GNB) when GNB assumes that all classes share a covariance matrix.

We used data from the calibration task to fit the BCI classifier. Input features (firing rate vectors) were calculated by counting the number of threshold crossings on each electrode during a 1-s time window within each trial's movement execution phase. The exact time window was a hyperparameter for each session and was chosen to maximize the cross-validated accuracy on the calibration dataset. To prevent low-firing rate discretization effects, we excluded electrodes with mean firing rates less than 1 Hz. This classifier was then used in subsequent online BCI control for the main task (finger flexion grid).

During online control of the finger flexion grid task, input features were similarly constructed by counting the threshold crossings from each electrode in a 1-s time window. This time window was fixed to [0.5, 1.5] s after cue presentation. The window start time was chosen based on the estimated saccade latency in the first experimental session. The saccade latency was estimated by taking the median latency for the subject to look >80% of the distance between targets. The analysis window was a priori determined to be 1 s; this choice was supported post hoc by a sliding window analysis (not shown), which confirmed that finger movements could be accurately classified up to 1.6 s after cue. The online classifier was occasionally retrained using data from this main task, usually every four run blocks.

Offline classification accuracy (*Figure 1—figure supplement 4*) was computed using leave-one-out cross-validation (within each session). We used features from the same time window as the online-control task. However, offline analyses used firing rates after spike-sorting, instead of raw threshold crossings.

In the Results, reported classification accuracies aggregate trials over all sessions (as opposed to averaging the accuracies across sessions with different numbers of trials). Reported SDs indicate variability across sessions, weighted by the number of trials in each session. To visualize confusion matrices, trials were pooled across sessions. Confusion matrix counts were normalized by row sum (true label) to display confusion percentages.

In the first session (2018-09-10), the No-Go condition (X) was not cued in the calibration task, so the classifier did not output No-Go predictions during that session. However, No-Go was cued in the main task; these 84 No-Go trials were thus excluded from the online-control accuracy metrics (*Figure 1b* and *Figure 1—figure supplement 3*), but they were included in the offline cross-validated confusion matrix (*Figure 1—figure supplement 4*).

## Cross-validated neural distance

We quantified the dissimilarity between the neural activity patterns of each finger pair $(j, k)$, using the cross-validated (squared) Mahalanobis distance (*Schütt et al., 2019*; *Nili et al., 2014*):

$$d_{jk}^2 = \left(b_j - b_k\right)_A \left(\frac{\Sigma_A + \Sigma_B}{2}\right)^{-1} \left(b_j - b_k\right)_B^T \, / \, N$$

where $A$ and $B$ denote independent partitions of the trials, $\Sigma$ are the partition-specific noise covariance matrices, $(b_j, b_k)$ are the trial measurements of firing rate vectors for conditions $(j, k)$, and $N$ normalizes for the number of neurons. The units of $d_{jk}^2$ are *unitless$^2$/neuron*.

The cross-validated Mahalanobis distance, also referred to as the 'crossnobis' distance (*Schütt et al., 2019*), measures the separability of multivariate patterns, analogous to LDA classification accuracy (*Nili et al., 2014*). To generate independent partitions $A$ and $B$ for each session, we stratified-split the trials into five non-overlapping subsets. We then calculated the crossnobis distance for each possible combination of subsets $(A, B)$ and averaged the results. Cross-validation ensures that the (squared) distance estimate is unbiased; $E\left[d_{jk}^2\right] = 0$ when the underlying distributions are identical (*Walther et al., 2016*). The noise covariance $\Sigma$ was regularized (*Ledoit and Wolf, 2003*) to guarantee invertibility.

Similar results were also obtained when estimating neural distances with the cross-validated Poisson symmetrized KL-divergence (*Schütt et al., 2019*; *Figure 2—figure supplement 3*).

## Representational models

We used RDMs to describe both the type and format of information encoded in a recorded population. To make these RDMs, we calculated the distances between each pair of finger movements and organized the 10 unique inter-finger distances into a $[n_{fingers}, n_{fingers}]$-sized RDM (*Figure 2d*).

Conveniently, the RDM abstracts away the underlying feature types, enabling direct comparison with RDMs across brain regions (*Kietzmann et al., 2019*), subjects, or recording modalities (*Kriegeskorte et al., 2008b*).

We also used RDMs to quantify hypotheses about how the brain might represent different actions. In *Figure 2b*, we generated an able-bodied model RDM using fMRI data from two independent studies, *Kieliba et al., 2021* (*N* = 29, pre-intervention, right hand, 3T scans) and *Ejaz et al., 2015* (*N* = 7, no intervention, right hand, 7T scans). The fMRI ROI was selected to match participant NS's anatomical implant location (PC-IP). Specifically, a 4-mm geodesic distance around vertex 7123 was initially drawn in fs_LR_32k space, then resampled onto fsaverage. The RDM for each subject was then calculated using the cross-validated (squared) Mahalanobis distance between fMRI activity patterns. Based on a permutation shuffle test, RDMs were similar between the studies' groups of participants, so we aggregated the RDMs into a single dataset here. The MC RDMs (*Figure 2—figure supplement 1*) used data from the same scans (*Ejaz et al., 2015*; *Kieliba et al., 2021*), with ROIs covering Brodmann area 4 near the hand knob of the precentral gyrus.

In *Figure 4* and its supplemental figures, we decomposed the data RDMs into model RDMs borrowed from *Ejaz et al., 2015*. The hand-usage model was constructed using the velocity time series of each finger's MCP joint during everyday tasks (*Ingram et al., 2008*). The muscle activity model was constructed using EMG activity during single- and multi-finger tasks. The somatotopic model is based on a cortical sheet analogy and assumes that finger activation patterns are linearly spaced Gaussian kernels across the cortical sheet. Further modeling details are available in the methods section of *Ejaz et al., 2015*.

## Comparing representational structures

We used the rsatoolbox Python library (*Schütt et al., 2019*) to calculate data RDMs and compare them with model RDMs (RSA) (*Kriegeskorte et al., 2008a*).

To quantify model fit, we used the whitened unbiased RDM cosine similarity (WUC) metric (*Diedrichsen et al., 2021*), which (*Diedrichsen et al., 2021*) recommend for models that predict continuous real values. We used WUC instead of Pearson correlation for two reasons (*Diedrichsen et al., 2021*). First, cosine similarity metrics like WUC properly exploit the informative zero point; because we used an unbiased distance estimate, $d_{jk}^2 = 0$ indicates that the distributions $(j, k)$ are identical. Second, Pearson correlation assumes that observations are independent, but the elements of each RDM covary (*Diedrichsen et al., 2021*) because the underlying dataset is shared. For example, the (thumb, middle)-pairwise dissimilarity uses the same thumb data as the (thumb, ring)-pairwise dissimilarity.

Like correlation similarities, a larger WUC indicates a better match, and the maximum WUC value is 1. However, cosine similarities like WUC are often larger than the corresponding correlation values or are even close to 1 (*Diedrichsen et al., 2021*). Thus, while correlation values can be compared against a null hypothesis of 0-correlation, WUC values should be interpreted by comparing against a baseline. The baseline is usually (*Diedrichsen et al., 2021*) chosen to be a null model where all conditions are pairwise-equidistant (and would thus correspond to 0-correlation). In this study, this happens to correspond to the unstructured model. For more details about interpreting the WUC metric, see *Diedrichsen et al., 2021*.

To demonstrate that our model comparisons were robust to the specific choice of RDM similarity metric, we also show model fits using whitened Pearson correlation in *Figure 2—figure supplement 3*. Whitened Pearson correlation is a common alternative to WUC (*Diedrichsen et al., 2021*).

## Noise ceiling for model fits

Measurement noise and behavioral variability cause data RDMs to vary across repetitions, so even a perfect model RDM would not achieve a WUC similarity of 1. To estimate the noise ceiling (*Nili et al., 2014*) (the maximum similarity possible given the observed variability between data RDMs), we assume that the unknown, perfect model resembles the average RDM. Specifically, we calculated the average similarity of each individual-session RDM (*Figure 2—figure supplement 2*) with the mean RDM across all other sessions (i.e., excluding that session):

$$\hat{C} = \frac{1}{D} \sum_{d=1}^{D} \text{similarity}(r_d, \bar{r}_{j} \neq d)$$

$$\bar{r}_{j \neq d} = \frac{1}{D-1} \sum_{j \neq d} r_j$$

where similarity is the WUC similarity function, $D$ is the number of RDMs, $r_d$ refers to a single RDM from an individual session, and $\hat{C}$ is the 'lower' noise ceiling. This noise ceiling is analogous to leave-one-out-cross-validation. If a model achieves the noise ceiling, the model fits the data well (*Nili et al., 2014*).

## Measuring changes in the representational structure

To assess the effect of BCI task experience on the inter-finger distances, we performed a linear regression analysis (*Figure 3b* and *Figure 3—figure supplement 1*). We first subdivided each session's dataset into individual runs and calculated separate RDMs for each (session, run) index. We then used linear regression to predict each RDM's (squared) inter-finger distances from the session index, run index, and finger pair:

$$d_{jk}^2 = \beta_{jk} + \beta_{session}s + \beta_{run}r + \beta_0$$

where $\beta_0$ is the average inter-finger distance, $\beta_{jk}$ is the coefficient for finger-pair $(j, k)$, $s$ is the session index, and $r$ is the run index. $|\beta_{session}| > 0$ would suggest that RDMs are dependent on experience across sessions. $|\beta_{run}| > 0$ would suggest that RDMs depend on experience across runs within a session. For *t*-tests, we conservatively estimated the degrees-of-freedom as the number of RDMs, because the individual elements of each RDM covary and thus are not independent (*Diedrichsen et al., 2021*). The effect sizes for the session-index predictor and the run-index predictor were quantified using *Cohen's $f^2$* (*Cohen, 1988*), comparing against the finger-pair-only model as a baseline.

For *t*-tests without significant differences, we also calculated BFs to determine the likelihood of the null hypothesis, using the common threshold that BF <1/3 substantially supports the null hypothesis (*Dienes, 2014*). BFs were computed using the R package BayesFactor (*Morey et al., 2015*) with default priors. To calculate BFs for one-sided *t*-tests (for example, $\beta > 0$), we sampled ($N = 10^6$) from the posterior of the corresponding two-sided *t*-test ($|\beta| > 0$), calculated the proportion of samples that satisfied the one-sided inequality, and divided by the prior odds (*Morey and Wagenmakers, 2014*) $(\frac{P(\beta > 0)}{P(|\beta| > 0)} = \frac{1}{2})$.

## Linear combinations of models

We modeled the finger RDM (in vector form) as a zero-intercept, non-negative linear combination (*Jozwik et al., 2016*) of potentially predictive model RDMs: usage, muscle, and somatomorphic (*Figure 4*).

First, we used the VIF to assess multicollinearity between the hypothesized models. For each model (e.g., usage), we calculated the standard, ordinary least squares (OLS)-based VIF (VIF$_{usage,OLS}$), and we also calculated a modified VIF (VIF$_{usage,NNLS}$) based on non-negative least squares (NNLS).

$$\text{VIF}_{j,\text{OLS}} = \frac{1}{1 - R^2_{M_j|M_{-j}}}$$

where $R^2_{M_j|M_{-j}}$ is the $R^2$ from an OLS regression predicting RDM $M_j$ from all other RDMs $M_{-j}$. VIF$_{OLS}$ values above a threshold indicate that multicollinearity is a problem; VIF >5 or VIF >10 are common thresholds (*James et al., 2013*). Here, we constrained the linear combination coefficients to be non-negative, which can sometimes mitigate multicollinearity. Thus, we also calculated VIF$_{NNLS}$, which follows the same equation above, except that we use NNLS to predict $M_j$ from $M_{-j}$.

Because multicollinearity was a problem here, we next determined the best subset of model RDMs to use. We used NNLS to predict the data RDM from the model RDMs. We estimated the model fits using leave-one-session-out cross-validation. To estimate model-fit uncertainty, we bootstrapped RDMs (sessions) over the cross-validation procedure (*Schütt et al., 2019*). We then used the 'one-standard error' rule (*James et al., 2013*) to select the best parsimonious model, choosing the simplest model within one standard error of the best model fit.

## Representational dynamics analysis

To investigate how the finger movement representational structure unfolds over time, we used a time-resolved version of RSA (*Kietzmann et al., 2019*; *Figure 4a*). At each timepoint within a trial, we computed the instantaneous firing rates by binning the spikes in a 200-ms time window centered at that point. These firing rates were used to calculate cross-validated Mahalanobis distances between each pair of fingers and generate an RDM. Snapshots (*Figure 4e*) show single-timepoint RDMs averaged across sessions.

The temporal sequence of RDMs constitutes an RDM movie (size $\left[n_{fingers}, n_{fingers}, n_{timepoints}\right]$) that visualizes the representational trajectory across the trial duration. RDM movies were computed separately for each recording session. At each timepoint, we linearly decomposed the data RDM into the component models using non-negative least squares. Because the component models were multi-collinear, component models were limited to the subsets chosen in the previous model reduction step. Each component RDM was normalized by its vector length ($\ell_2$-norm) before decomposition to allow comparison between coefficient magnitudes. We used bootstrapped sampling of RDMs across sessions and decomposed the bootstrap-mean RDM to generate a confidence intervals on the coefficients.

We computed the start time of each model component as the time at which the corresponding mixture coefficient exceeded 0.2 (about 25% of the median peak coefficient across models and sessions).

## Acknowledgements

We thank NS for her dedicated participation in the study. Kelsie Pejsa and Viktor Scherbatyuk for administrative and technical assistance. Paulina Kieliba and Tamar Makin for sharing their fMRI data. Tamar Makin and Whitney Griggs for their helpful feedback on the manuscript. Jörn Diedrichsen and Spencer Arbuckle for sharing their fMRI data and models.

## Additional information

### Funding

| Funder | Grant reference number | Author |
| --- | --- | --- |
| National Eye Institute | R01EY015545 | Charles Guan<br>Richard A Andersen |
| National Eye Institute | UG1EY032039 | Charles Guan<br>Richard A Andersen |
| Tianqiao and Chrissy Chen Brain-machine Interface Center at Caltech | | Tyson Aflalo<br>Richard A Andersen |
| Boswell Foundation | | Richard A Andersen |
| Amazon AI4Science Fellowship | | Charles Guan |

The funders had no role in study design, data collection, and interpretation, or the decision to submit the work for publication.

### Author contributions

Charles Guan, Conceptualization, Data curation, Software, Formal analysis, Funding acquisition, Validation, Investigation, Visualization, Methodology, Writing – original draft, Writing – review and editing; Tyson Aflalo, Conceptualization, Formal analysis, Supervision, Funding acquisition, Validation, Investigation, Methodology, Writing – original draft, Writing – review and editing; Carey Y Zhang, Software, Investigation; Elena Amoruso, Resources, Data curation, Formal analysis, Funding acquisition, Project administration; Emily R Rosario, Resources, Funding acquisition, Methodology, Project administration; Nader Pouratian, Resources, Supervision, Funding acquisition, Validation, Methodology, Project administration, Writing – review and editing, surgery to implant the recording arrays;

Richard A Andersen, Resources, Data curation, Formal analysis, Supervision, Funding acquisition, Validation, Project administration, Writing – review and editing

**Author ORCIDs**
Charles Guan ![ORCID] http://orcid.org/0000-0002-8040-8844
Tyson Aflalo ![ORCID] http://orcid.org/0000-0002-0101-2455
Emily R Rosario ![ORCID] http://orcid.org/0000-0002-1540-197X
Nader Pouratian ![ORCID] http://orcid.org/0000-0002-0426-3241
Richard A Andersen ![ORCID] http://orcid.org/0000-0002-7947-0472

**Ethics**
Clinical trial registration ClinicalTrials.gov identifier: NCT01958086.
All procedures were approved by the California Institute of Technology, Casa Colina Hospital and Centers for Healthcare, and the University of California, Los Angeles Institutional Review Boards. NS consented to the surgical procedure as well as to the subsequent clinical studies after understanding their nature, objectives, and potential risks.

**Decision letter and Author response**
Decision letter https://doi.org/10.7554/eLife.74478.sa1
Author response https://doi.org/10.7554/eLife.74478.sa2

## Additional files

**Supplementary files**
• Transparent reporting form

**Data availability**
Data are available on the BRAIN Initiative DANDI Archive at https://dandiarchive.org/dandiset/000147. Code to reproduce the main figures is available at: https://github.com/AndersenLab-Caltech/fingers_rsa (copy archived at swh:1:rev:6915bf863ad1339e6fc8c6886032cd0f70fd8b10).

The following dataset was generated:

| Author(s) | Year | Dataset title | Dataset URL | Database and Identifier |
|---|---|---|---|---|
| Guan C, Aflalo T, Zhang C, Andersen RA | 2022 | PPC_Finger: human posterior parietal cortex recordings during attempted finger movements | https://dandiarchive.org/dandiset/000147 | DANDI, 000147 |

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
