## [Editor Report]

Using data from an tetraplegic individual, the authors show that the neural representations for attempted single finger movements after multiple years after the injury is still organized in a way that is typical for healthy participants. They also show that the representational structure does not change during task training on a simple finger classification task – and that the representational structure, even without active motor outflow or sensory inflow, switches from a motor representation to a sensory representation during the trial. The analyses are convincing, and the results have important implications for the use and training of BCI devices in humans.

---

## [Decision Letter]

**Decision letter after peer review:**

Thank you for submitting your article "Preserved motor representations after paralysis" for consideration by *eLife*. Your article has been reviewed by 3 peer reviewers, including Jörn Diedrichsen as Reviewing Editor and Reviewer #1, and the evaluation has been overseen by Richard Ivry as the Senior Editor.

Essential revisions:

1) One important question to address clearly in the revision is the mismatch of the representational structure of the recorded PPC site with the SPLa ROI from the imaging studies (and the match with M1). Reviewer #1 raises the important questions that fMRI ROI is likely not equivalent to the recorded site and should be modified and the finding reinvestigated. If the discrepancy persists after the reanalysis, Reviewer #2, and #3 make suggestions for a clearer discussion of the findings. Overall, the claim of the stability of pre-injury organization probably needs to be tempered a bit.

2) The stability of representations across learning is an interesting finding, but Reviewer #2 and #3 both made the valid request that you should more clearly discuss possible reasons for why the basic manifold of neural activity was preserved, by comparing it to previous studies that have found "off-manifold" learning (delayed feedback, recalibration, training protocol – see reviews). In this context you should also discuss the limitations of having N=1. Although the reviewers agree that case studies in this scenario can be valuable, the limitations in generalization should be noted and discussed adequately.

3) All reviewers agreed also that the temporal change of the representational structure was a very interesting point. Please address the specific comments raised in the reviews for this analysis. As this is likely the most novel contribution of the paper, we believe it should be more explicitly mentioned in the abstract / introduction and more clearly discussed at the end of the paper.

*Reviewer #1 (Recommendations for the authors):*

I found the investigation very interesting and illuminating. Personally, I believe that the third result (i.e., the time-dependency of the representational structure) is one of the most interesting results of the paper, and should be mentioned in the abstract / introduction.

My only main comment concerns the first result – the comparison of the representational structure with previous fMRI results. Judging from Figure 1 – supp 1, the implantation site is located right on the boundary between the postcentral gyrus and posterior parietal cortex, seemingly ~5mm anterior to the peak of activation observed during grasping (AIP). Thus, based on the ROI definition used in both Ejaz et al., (2015) and Kieliba at al., (2021), it would be right on the boundary between SPLa and S1 – and cycto-architectonically, right on the boundary between BA2 and BA5. The SPLa ROI in Kieliba at al. extends substantially posterior to this. Thus, it is not clear whether the observed mismatch between neuronal recordings and fMRI reveals a real discrepancy (see Arbuckle et al., 2020), a function of the low reliability of the SPLa RDM (as discussed), or the mismatch in location. To improve this analysis, I would be happy to provide an RD from a more tightly matched ROI from the Ejaz et al., (2015) data, if needed.

*Reviewer #2 (Recommendations for the authors):*

Suggestions:

1) As described in the public review, the claim that motor representations are preserved after paralysis needs to be revised and described with more nuance to reflect the complex results that were presented in the manuscript. These revisions should be made to the title and throughout the discussion.

2) The third hypothesis (that pre-injury motor representations could have de-specialized) references Figure 2C, but that same figure panel was referenced for the second hypothesis (that the PC-IP representation would match that from SPLa in the imaging studies). It seems that the third hypothesis is not represented in Figure 2 so it should be added.

3) In figure 3a, it does appear that the single unit model became slightly less similar to the M1 model over time and in Figure 3c, the MDS plot suggests that the activity associated with each finger became slightly more variable (larger standard deviation) in the later sessions. While not a dramatic effect, could this be indicative of learning? The BCI performance should be shown for each session. Was there any trend in the overall success rate or error pattern over time?

4) The somatotopic model fit appeared to be stronger (in terms of model coefficient) than the muscle or usage model. Is this surprising given the interpretation that the somatotopic representation is primarily driven by sensory feedback, and this is a covert task. Might one expect the feedforward efferent signal to be stronger? Further, the discussion mentions that there is no topography in PPC, These seem to be in conflict. This may require a more nuanced discussion of the limitations of fMRI temporal resolution that may impact the ability to measure a consistent pattern. Further, the authors reference a previous study, Chivukula et al., as evidence of touch representations in PPC.

5) The discussion about limitations in the learning aspects of the study should be expanded as described in the weaknesses section of the public review. In addition, please comment on how the choice of the decoding window and features should impact the results given that there appears to be two temporally distinct representations measured in parietal cortex. Perhaps the participant converged on a strategy that was essentially a compromise between these two patterns, which both would have been included during the decoding window. Nevertheless, that no learning was observed in the current study is an important finding worthy of dissemination.

*Reviewer #3 (Recommendations for the authors):*

I'll avoid repeating concerns from the public review; most of those can be addressed by adding additional caveats or changing the introduction framing. As just one specific example, the manuscript would thus be substantially improved by appropriately narrowing the scope of claims. E.g., the last line of the Introduction would be only marginally longer, but would much better reflect the overall state of knowledge, if it were edited to say "Our results reveal that adult motor representations are preserved **in PPC** even after years without use".

Page 26 line 481: I really didn't understand this paragraph about estimating noise ceiling (and thus also I struggled to assess the claim that the model fit was "close to the theoretical maximum" on page 8 line 125). I understand at a high level what you're trying to do, but there are a lot of ways to estimate what even a perfect fit would look like given the single trial / spike measurement noise present. Can you explain the motivation/specific approach more (what do you mean by noise of the RDM?) and in more detail explain this calculation? Both more explanatory text, and adding the equations to Methods, would help. In general I found that the Methods could be more clear throughout and would recommend asking a colleague in a different systems neuroscience lab to read the Methods and explain what you did back to you (to reveal places where a reader would not be able to replicate the steps).

Page 26 line 491: Can you first briefly explain what the "directional hypothesis" is? Or else the reader would have had to read the Ejaz et al., methods to follow this section.

Can authors explain how hypothesis #1 (RSA structure matches natural statistics of use, as per ref 14) is distinct from hypothesis #2 (the BCI finger representation in PC-IP might instead match the representation of able-bodied individuals in the same brain area")? Couldn't #1 and #2 describe the same RSA structure (it seems that in fact they are highly correlated)? Was there past work showing they are different? Is there a reference to help unpack hypothesis #2? Right now hypothesis #1 vs #2 are really just illustrated by the two different RSA matrices in Figure 2b vs 2c, but that's asking the reader to squint and judge how different those matrices are. Some more hand-holding up front of where these hypotheses differ (and why) would help set the stage for this key question and result.

"In the first human BCI study of neural-population plasticity" (page 18, line 330) is a questionable claim (and unnecessarily provocative, in my view). For example, Wodlinger et al., (2015) examined BCI control of a robot arm over many sessions across ~250 days of use (Figure 5, and "showing that learning took place consistently over a long period of time") – this would seem to be an arm BCI learning study (asking, does neural activity change in a way that would lead to improved performance over time, by examining the decoder output which is, by definition, the task-relevant readout of the motor cortical population). That is not that different from how this study examines finger BCI learning. Another example is Eichenlaub, Jarosiewicz et al., (2020), which looked at learning-associated replay events in a BCI-controlled sequence task.

The page 18 line 331-332 sentence (about handwriting) also puzzled me: didn't that study explicitly find that yes, handwriting remains preserved through at least a decade of injury? Perhaps the present manuscript can be more specific about what future studies should ask about these complex motor skills

---

## [Author Response]

Essential revisions:1) One important question to address clearly in the revision is the mismatch of the representational structure of the recorded PPC site with the SPLa ROI from the imaging studies (and the match with M1). Reviewer #1 raises the important questions that fMRI ROI is likely not equivalent to the recorded site and should be modified and the finding reinvestigated.

In the original submission, we found that PC-IP single neurons more strongly matched the MC fMRI RDMs than the SPLa fMRI RDMs, as measured by the whitened unbiased cosine (WUC) similarity (Diedrichsen et al., 2021). After narrowing the SPLa fMRI ROI to the implant region (PC-IP), a similar pattern of results still held. The PC-IP fMRI RDM matched the tetraplegic participant's data better than the task-optimal, unstructured model. Furthermore, participant NS's finger organization still matched the MC fMRI data more strongly than the PC-IP fMRI data from the same participants.

If the discrepancy persists after the reanalysis, Reviewer #2, and #3 make suggestions for a clearer discussion of the findings. Overall, the claim of the stability of pre-injury organization probably needs to be tempered a bit.

Given this discrepancy, we have tempered the preservation claims in the manuscript; we changed the title from "Preserved motor representations after paralysis" to "Stability of motor representations after paralysis." We comment on possible causes for the discrepancy as well:

"This result does obscure a straightforward interpretation of the RSA results – why does our recording area match MC better than the corresponding implant location? Several factors might contribute, including differing neurovascular sensitivity to the early and late phases of the neural response (Figure 4e), heterogeneous neural organizations across the single-neuron and voxel spatial scales (Arbuckle et al., 2020; Guest and Love, 2017; Kriegeskorte and Diedrichsen, 2016), or mismatches in functional anatomy between participant NS and standard atlases (Eickhoff et al., 2018)."

Still, we find it compelling that participant NS's finger representation strongly matches the MC fMRI RDM, which is known to reflect the natural hand use patterns of able-bodied individuals (Ejaz et al., 2015). Participant NS has not moved her own hands in 10 years, and BCIs remove any biomechanical constraints that might encourage correlated representations.

Revised text in the Discussion:

"Despite the lack of overt movement or biomechanical constraints, the measured finger representation still reflected these usage-related patterns."

Based on our RSA and learning analyses (Figures 2 and 3), such a structure is unlikely to have emerged after paralysis. The apparent preservation of the forward model (Figure 4e), as well as other recent studies of tetraplegic individuals, also corroborate an interpretation that motor representations are preserved. However, we acknowledge that single-neuron recordings in participants would provide the most direct comparison baseline, although these recordings are difficult to acquire.

Revised text in the Discussion:

"Comparisons with single-neuron recordings from able-bodied participants would validate this interpretation but may be difficult to acquire."

2) The stability of representations across learning is an interesting finding, but Reviewer #2 and #3 both made the valid request that you should more clearly discuss possible reasons for why the basic manifold of neural activity was preserved, by comparing it to previous studies that have found "off-manifold" learning (delayed feedback, recalibration, training protocol – see reviews). In this context you should also discuss the limitations of having N=1. Although the reviewers agree that case studies in this scenario can be valuable, the limitations in generalization should be noted and discussed adequately.

We agree with the reviewers that a different experimental paradigm could encourage off-manifold learning. We have expanded the discussion to clarify this point.

Revised text in the Discussion:

"The stability of finger representations here suggests that BCIs can benefit from the pre-existing, natural repertoire (Hwang et al., 2013), although learning can play an important role under different experimental constraints. In our study, the participant received only a delayed, discrete feedback signal after classification (Figure 1a). Because we were interested in understanding participant NS’s natural finger representation, we did not artificially perturb the BCI mapping. However, a previous study showed that BCI users can slowly learn to generate new, off-manifold neural activity patterns when the user receives continuous feedback and the BCI decoder is perturbed incrementally (Oby et al., 2019). Notably, learning was inconsistent when perturbations were sudden, indicating that learning is sensitive to specific guiding procedures. To further understand how much finger representations can be actively modified, future studies could benefit from perturbations (Kieliba et al., 2021; Oby et al., 2019), richer neurofeedback, and additional participants. "

2) The stability of representations across learning is an interesting finding, but Reviewer #2 and #3 both made the valid request that you should more clearly discuss possible reasons for why the basic manifold of neural activity was preserved, by comparing it to previous studies that have found "off-manifold" learning (delayed feedback, recalibration, training protocol – see reviews). In this context you should also discuss the limitations of having N=1. Although the reviewers agree that case studies in this scenario can be valuable, the limitations in generalization should be noted and discussed adequately.3) All reviewers agreed also that the temporal change of the representational structure was a very interesting point. Please address the specific comments raised in the reviews for this analysis. As this is likely the most novel contribution of the paper, we believe it should be more explicitly mentioned in the abstract / introduction and more clearly discussed at the end of the paper.

We have expanded the abstract and introduction to further highlight the temporal analysis.

Revised text in the Abstract:

"Within individual BCI movements, the representational structure was dynamic, first resembling muscle activation patterns and then resembling the anticipated sensory consequences."

Revised text in the Introduction:

"Temporal dynamics provide another lens to investigate neural organization and its changes after paralysis. Temporal signatures can improve BCI classification (Willett et al., 2021) or provide a baseline for motor adaptation studies (Stavisky et al., 2017; Vyas et al., 2018). Notably, motor cortex activity exhibits quasi-oscillatory dynamics during arm reaching (Churchland et al., 2012). More generally, the temporal structure can depend on the movement type (Suresh et al., 2020) and the recorded brain region (Schaffelhofer and Scherberger, 2016). In this study, we recorded from the posterior parietal cortex (PPC), which is thought to compute an internal forward model for sensorimotor control (Desmurget and Grafton, 2000; Li et al., 2022; Mulliken et al., 2008; Wolpert et al., 1998). A forward model overcomes inherent sensory delays to enable fast control by predicting the upcoming states. If PPC activity resembles a forward model even after paralysis, this would suggest that even the temporal details of movement are preserved after injury."

“Furthermore, the neural representational dynamics matched the temporal profile expected of a forward model in able-bodied individuals, first resembling muscle activation patterns and then resembling expected sensory outcomes.”

Reviewer #1 (Recommendations for the authors):I found the investigation very interesting and illuminating. Personally, I believe that the third result (i.e., the time-dependency of the representational structure) is one of the most interesting results of the paper, and should be mentioned in the abstract / introduction.

We have expanded the abstract and introduction to include this, as mentioned above in the response to the review summary.

My only main comment concerns the first result – the comparison of the representational structure with previous fMRI results. Judging from Figure 1 – supp 1, the implantation site is located right on the boundary between the postcentral gyrus and posterior parietal cortex, seemingly ~5mm anterior to the peak of activation observed during grasping (AIP). Thus, based on the ROI definition used in both Ejaz et al., (2015) and Kieliba at al. (2021), it would be right on the boundary between SPLa and S1 – and cycto-architectonically, right on the boundary between BA2 and BA5. The SPLa ROI in Kieliba at al. extends substantially posterior to this. Thus, it is not clear whether the observed mismatch between neuronal recordings and fMRI reveals a real discrepancy (see Arbuckle et al., 2020), a function of the low reliability of the SPLa RDM (as discussed), or the mismatch in location. To improve this analysis, I would be happy to provide an RD from a more tightly matched ROI from the Ejaz et al., (2015) data, if needed.

We have updated the analyses using anatomically matched fMRI data from Reviewer #1 (Jorn Diedrichsen) and from our original collaborator (Elena Amoruso).

Reviewer #2 (Recommendations for the authors):Suggestions:1) As described in the public review, the claim that motor representations are preserved after paralysis needs to be revised and described with more nuance to reflect the complex results that were presented in the manuscript. These revisions should be made to the title and throughout the discussion.

We have updated the manuscript as per the suggestion and as described above, under essential revisions.

2) The third hypothesis (that pre-injury motor representations could have de-specialized) references Figure 2C, but that same figure panel was referenced for the second hypothesis (that the PC-IP representation would match that from SPLa in the imaging studies). It seems that the third hypothesis is not represented in Figure 2 so it should be added.

This was an error in the subfigure indexing and has now been updated.

3) In figure 3a, it does appear that the single unit model became slightly less similar to the M1 model over time and in Figure 3c, the MDS plot suggests that the activity associated with each finger became slightly more variable (larger standard deviation) in the later sessions. While not a dramatic effect, could this be indicative of learning?

Overall, the decreases in distances contradict the expected learning hypothesis that participant NS would increase distances to perform the BCI task better. It appears that the decrease in inter-finger distances, while significant, was small enough to not affect classification.

Looking at the specific figures mentioned, the model fit metric (WUC similarity) used in Figure 3a requires a baseline to be compared against (Diedrichsen et al., 2021). Thus, the appropriate statistical test is to compare the slope of the fMRI model’s fit-values against the slope of the Unstructured model’s fit-values. The statistical tests do not find a difference in slopes, although we acknowledge that a small, significant trend could emerge if participant NS had participated in even more sessions.

The inter-finger distances decrease slightly in later sessions (Figure 3c, closer points in the MDS plot; Figure 3-—figure supplement 1, negative slope), although the effect size is rather small. As lower distances would generally indicate worse classification, this also runs counter to the learning hypothesis, which expects that participant NS should increase the neural distances to perform the BCI control task better.

Is it possible that learning took some form other than increased neural distances? One option would be that participant NS learned to control her neural variance; i.e. her attempted movements were more stereotyped, even, for example, as the means of her ring and pinky fingers moved closer together. However, the crossnobis (cross-validated Mahalanobis) distance already normalizes by the neural variance. In other words, lower-variance distributions appear farther away from each other.

The BCI performance should be shown for each session. Was there any trend in the overall success rate or error pattern over time?

We have added the BCI performance for each session in the new Figure 1—figure supplement 3. The evidence was not strong for an across-session trend in the online BCI accuracy (P = 0.060, BF = 1.7, slope = 0.009 / session).

4) The somatotopic model fit appeared to be stronger (in terms of model coefficient) than the muscle or usage model. Is this surprising given the interpretation that the somatotopic representation is primarily driven by sensory feedback, and this is a covert task. Might one expect the feedforward efferent signal to be stronger?

This is an interesting question. First, to clarify, we do not think that the somatotopic representation is driven by sensory feedback, but instead, represents the internal state estimate of the limb. State estimates, or internal abstractions that encode the brain’s estimate of the current position and velocity of the limb, are needed as sensory feedback is delayed, noisy, and multimodal. In motor intact individuals, this state estimate would combine sensory inputs with an internally produced estimate of the state as derived from known motor commands and a model of the body. As you mention, a loss of sensory inputs could potentially lead to a massive loss of neural representation. However, we know from the phantom limb literature that individuals can maintain internal representations of their lost limbs that they can still manipulate in their minds eye. This supports the idea that individuals maintain strong internal models of the body, even after catastrophic injury. Based on this prior evidence, it is not overly surprising that there exists a strong somatotopy representation. In terms of the relative strength of the two signals, we speculate that this will largely depend on brain region, e.g., we can imagine a region that represents the computed state estimate without any muscle-like model.

Further, the discussion mentions that there is no topography in PPC, These seem to be in conflict. This may require a more nuanced discussion of the limitations of fMRI temporal resolution that may impact the ability to measure a consistent pattern. Further, the authors reference a previous study, Chivukula et al., as evidence of touch representations in PPC.

Here it is important to distinguish cortical topography (e.g. a spatially organized mapping of the body onto the 2D cortical sheet, such as the homunculus) from a somatotopic representation (e.g., firing of single neurons as a smooth parametric function of body location.) Somatotopic representations can occur without a cortical topography if the somatotopic neurons are randomly distributed across the 2D cortical surface. This is consistent with our data in that representations of individual digits are all randomly intermingled within the same 4x4 mm patch of cortex that is sampled by our electrode array. Likewise, in Chivukula et al., tactile fields for all sensate parts of the body were also randomly intermingled in the 4x4 mm patch of cortex. Thus, even at the single neuron scale, we see no cortical topography, which may help to explain why no topography exists in the fMRI literature within PPC.

5) The discussion about limitations in the learning aspects of the study should be expanded as described in the weaknesses section of the public review.

This has been updated, as described above in essential revisions.

In addition, please comment on how the choice of the decoding window and features should impact the results given that there appears to be two temporally distinct representations measured in parietal cortex. Perhaps the participant converged on a strategy that was essentially a compromise between these two patterns, which both would have been included during the decoding window.

The input features were 1 firing rate/electrode, averaged over the 1-second decoding window. Knowing now that there appear to be two temporally distinct representations, it might have been interesting to split up the BCI decoding window into separate features through time. Perhaps this could help facilitate separate learning in the early and late components.

Revised text in the Discussion:

“Finally, given that finger representations were dynamic (Figure 4e), learning might occur separately in the early and late dynamics. Time-variant BCI decoding algorithms, such as recurrent neural networks (Sussillo et al., 2012; Willett et al., 2021), could help facilitate learning specific to different time windows of movement.”

Nevertheless, that no learning was observed in the current study is an important finding worthy of dissemination.Reviewer #3 (Recommendations for the authors):I'll avoid repeating concerns from the public review; most of those can be addressed by adding additional caveats or changing the introduction framing. As just one specific example, the manuscript would thus be substantially improved by appropriately narrowing the scope of claims. E.g., the last line of the Introduction would be only marginally longer, but would much better reflect the overall state of knowledge, if it were edited to say "Our results reveal that adult motor representations are preserved **in PPC** even after years without use".

This has been updated, as described above in essential revisions.

Page 26 line 481: I really didn't understand this paragraph about estimating noise ceiling (and thus also I struggled to assess the claim that the model fit was "close to the theoretical maximum" on page 8 line 125). I understand at a high level what you're trying to do, but there are a lot of ways to estimate what even a perfect fit would look like given the single trial / spike measurement noise present. Can you explain the motivation/specific approach more (what do you mean by noise of the RDM?) and in more detail explain this calculation? Both more explanatory text, and adding the equations to Methods, would help.

We agree that many different sources that could contribute to variability in neuroscience results. RDM noise ceilings usually only measure variability across RDMs (i.e., variability across sessions or subjects), not single-trial/spike measurement noise. Trial-to-trial variability (including spike measurement noise) is accounted for by the use of the cross-validated Mahalanobis distance, which normalizes single-neuron firing rates by an estimate of their trial-to-trial variability (i.e., noise covariance). Thus, we are mainly concerned with the variability across repeated sessions or subjects.

Revised Methods section:

“Noise ceiling for model fits

Measurement noise and behavioral variability cause data RDMs to vary across repetitions, so even a perfect model RDM would not achieve a WUC similarity of 1. To estimate the noise ceiling(Nili et al., 2014) (the maximum similarity possible given the observed variability between data RDMs), we assume that the unknown, perfect model resembles the average RDM. Specifically, we calculated the average similarity of each individual-session RDM (Figure 2—figure supplement 2) with the mean RDM across all other sessions (i.e., excluding that session):C^=1D∑d=1Dsimilarity(rd,r¯j≠d)r¯j≠d=1D−1∑j≠drj

where similarity is the WUC similarity function, D is the number of RDMs, rd refers to a single RDM from an individual session, and C^ is the “lower” noise ceiling. This noise ceiling is analogous to leave-one-out-cross-validation. If a model achieves the noise ceiling, the model fits the data well (Nili et al., 2014).”

In general I found that the Methods could be more clear throughout and would recommend asking a colleague in a different systems neuroscience lab to read the Methods and explain what you did back to you (to reveal places where a reader would not be able to replicate the steps).

Great suggestion. We followed this advice and hope the revised Methods are more clear.

Page 26 line 491: Can you first briefly explain what the "directional hypothesis" is? Or else the reader would have had to read the Ejaz et al., methods to follow this section.

We originally used the term “directional hypothesis” to refer to one-sided t-tests; we have revised our terminology to make this explicit.

Revised text in Methods:

“To calculate Bayes factors for one-sided t-tests (for example, β>0), we sampled (N = 106) from the posterior of the corresponding two-sided t-test (|β|>0), calculated the proportion of samples that satisfied the one-sided inequality, and divided by the prior odds (Morey and Wagenmakers, 2014) (P(β>0)P(|β|>0) = 12 ).”

Can authors explain how hypothesis #1 (RSA structure matches natural statistics of use, as per ref 14) is distinct from hypothesis #2 (the BCI finger representation in PC-IP might instead match the representation of able-bodied individuals in the same brain area”)? Couldn’t #1 and #2 describe the same RSA structure (it seems that in fact they are highly correlated)? Was there past work showing they are different? Is there a reference to help unpack hypothesis #2? Right now hypothesis #1 vs #2 are really just illustrated by the two different RSA matrices in Figure 2b vs 2c, but that’s asking the reader to squint and judge how different those matrices are. Some more hand-holding up front of where these hypotheses differ (and why) would help set the stage for this key question and result.

We agree that, a priori, it is possible hypotheses #1 and #2 could describe the same RSA structure. Indeed, at a high level, the PC-IP and motor cortex (MC) fMRI finger RDMs are more similar than they are different. To make this less confusing to readers, we have revised the way we introduce the hypotheses, deferring the previous hypothesis #1 to later.

Revised text in the Results:

“We used RSA to test three hypotheses: (1) the BCI finger representational structure could match that of able-bodied individuals (Ejaz et al., 2015; Kieliba et al., 2021) (Figure 2b and Figure2—figure supplement 1), which would imply that motor representations did not reorganize after paralysis… We note that our able-bodied model was recorded from human PC-IP using fMRI, which measures fundamentally different features (millimeter-scale blood oxygenation) than microelectrode arrays (sparse sampling of single neurons). Another possibility is that (2) the participant’s pre-injury motor representations had de-specialized after paralysis, such that finger activity patterns are unstructured and pairwise-independent…”

Only later do we compare to fMRI recordings from another region (i.e., MC). We also have revised the text to explain the quantitative differences between the PC-IP and MC fMRI finger RDMs.

Revised text in the Results:

“We also compared the PC-IP BCI RDM with able-bodied fMRI motor cortex (MC) RDMs, which have been previously shown to match the patterns of natural hand use (Ejaz et al., 2015). The able-bodied MC and PC-IP fMRI finger organizations are similar in that they represent the thumb distinctly from the other fingers, but PC-IP represents each of the non-thumb fingers similarly while MC distinguishes between all five fingers (Figure 2—figure supplement 1).”

Revised text in the Discussion:

“Compared to the PC-IP fMRI finger representation, MC represents the non-thumb fingers more distinctly from each other (Figure 2—figure supplement 1).”

“In the first human BCI study of neural-population plasticity” (page 18, line 330) is a questionable claim (and unnecessarily provocative, in my view). For example, Wodlinger et al., (2015) examined BCI control of a robot arm over many sessions across ~250 days of use (Figure 5, and "showing that learning took place consistently over a long period of time") – this would seem to be an arm BCI learning study (asking, does neural activity change in a way that would lead to improved performance over time, by examining the decoder output which is, by definition, the task-relevant readout of the motor cortical population). That is not that different from how this study examines finger BCI learning. Another example is Eichenlaub, Jarosiewicz et al., (2020), which looked at learning-associated replay events in a BCI-controlled sequence task.

Removed as recommended.

The page 18 line 331-332 sentence (about handwriting) also puzzled me: didn't that study explicitly find that yes, handwriting remains preserved through at least a decade of injury? Perhaps the present manuscript can be more specific about what future studies should ask about these complex motor skills

We have updated the discussion to clarify the point we were trying to make:

“As BCIs enable more complex motor skills, such as handwriting (Willett et al., 2021), future studies could investigate whether these complex skills also retain their pre-injury representational structure. For example, does a tetraplegic participant’s BCI handwriting look like their physical, pre-injury handwriting? These results will have important implications for the design of future neural prosthetics.”